# Effective Reasoning Chains Reduce Intrinsic Dimensionality

**Archiki Prasad**[* 1]   **Mandar Joshi**[2]   **Kenton Lee**[2]   **Mohit Bansal**[1]   **Peter Shaw**[2]

## Abstract

Chain-of-thought (CoT) reasoning and its variants have substantially improved the performance of language models on complex reasoning tasks, yet the precise mechanisms by which different strategies facilitate generalization remain poorly understood. While current explanations often point to increased test-time computation or structural guidance, establishing a consistent, quantifiable link between these factors and generalization remains challenging. In this work, we identify *intrinsic dimensionality* as a quantitative measure for characterizing the effectiveness of reasoning chains. Intrinsic dimensionality quantifies the minimum number of model dimensions needed to reach a given accuracy threshold on a given task. By keeping the model architecture fixed and varying the task formulation through different reasoning strategies, we demonstrate that effective reasoning strategies consistently reduce the intrinsic dimensionality of the task. Validating this on GSM8K with Gemma-3 1B and 4B, we observe a strong inverse correlation between the intrinsic dimensionality of a reasoning strategy and its generalization performance on both in-distribution and out-of-distribution data. Our findings suggest that effective reasoning chains facilitate learning by better compressing the task using fewer parameters, offering a new quantitative metric for analyzing reasoning processes.

## 1. Introduction

Chain-of-thought reasoning (CoT) – whether through few-shot prompting (Wei et al., 2022), zero-shot prompting (Kojima et al., 2022), or various post-training methods (Zelikman et al., 2022; Chung et al., 2024) – has substantially improved the performance of large language models (LLMs)

on reasoning tasks by generating textual rationales before final answers. Subsequent work has proposed numerous variations with different stylistic and strategic features, including code-based solutions (Gao et al., 2023; Chen et al., 2023), decomposition strategies (Zhou et al., 2023; Khot et al., 2023; Wang et al., 2023b), and extended reasoning with verification loops (Snell et al., 2024; Muennighoff et al., 2025). These variations represent different ways of communicating problem-solving strategies and structuring solutions – analogous to how humans adapt their communication style to their interlocutor in dialogue (Pickering & Garrod, 2004; Giles et al., 1991). Empirical evidence shows different reasoning strategies yield varying performance across tasks (Zhou et al., 2024), consistent with the intuition that different solution approaches suit different problems or learners. Further, not all problems benefit from generating rationales prior to the answer (Sprague et al., 2025).

This motivates an important research question: *when and why is reasoning effective, and given different reasoning strategies, which is most effective for improving model performance?* Existing explanations in prior work suffer from notable limitations. First, qualitative hypotheses about the importance of "structure" or relevance of a reasoning chains are not quantifiable (Wang et al., 2023a; Li et al., 2025). Consequently, these hypotheses are subject to interpretation, limiting both their predictive capacity and the ability to offer a theoretically grounded explanation for what makes reasoning effective. On the other hand, prevalent quantitative measures are often associated with conflicting evidence. For example, the relationship between the length of reasoning trajectories and the subsequent increased inference-time computational capacity remains unclear; while some works find clear gains (Muennighoff et al., 2025; Li et al., 2025), other work reports that shorter chains can be more effective and that continuing to extend reasoning (e.g., via "wait" tokens) can yield degradation in performance (Wu et al., 2026; Marjanović et al., 2025). Current approaches such as process reward models or correctness-based classifiers also require subjective specifications of desirable properties and do not provide a principled measure of effectiveness. A reliable quantitative measure would have significant practical implications: it could inform how to annotate or collect reasoning data, how to align reasoning strategies to particular student models, and how to design better regularizers

---
[1]UNC Chapel Hill [2]Google DeepMind. [*]*Work partially done during an internship at Google DeepMind*. Correspondence to: Archiki Prasad <archiki@cs.unc.edu>.

*Proceedings of the 43$^{rd}$ International Conference on Machine Learning*, Seoul, South Korea. PMLR 306, 2026. Copyright 2026 by the author(s).

that avoid limiting exploration or reward models grounded in generalization principles rather than subjective criteria.

To address this gap, we draw on the long-standing literature that uses information-theoretic perspectives to explain the efficacy of neural networks. Foundational concepts such as the minimum description length principle (Rissanen, 1978; Grünwald, 2007) posit an inverse relationship between the capacity required to represent a solution and its expected generalization. Building on this, the notion of intrinsic dimensionality (Li et al., 2018; Aghajanyan et al., 2021) applies these insights for overparameterized models, measuring the effective number of parameters needed to fit a given task objective. Specifically, intrinsic dimensionality is a function of both the model and the task. While prior work has typically fixed the data to analyze how different models vary in their intrinsic dimensionality, we instead fix the model and vary the training data by changing the reasoning strategy used to generate solutions. Although the underlying capability required (e.g., solving math problems) remains constant, different reasoning strategies change the supervision provided to the model during training. In this context, one might intuitively expect that requiring a model to generate long reasoning chains alongside final answers would increase the complexity of the outputs, making the task harder to fit. However, we hypothesize the opposite for *effective reasoning*: if a reasoning strategy effectively bridges the logical gap between input and answer, it should render the underlying mapping *more compressible*, requiring fewer degrees of freedom to learn, thereby resulting in *lower intrinsic dimensionality*.

We demonstrate a *strong inverse correlation between intrinsic dimensionality and generalization performance* across multiple chain-of-thought variants on GSM8K (Cobbe et al., 2021). These findings hold for both Gemma-3 1B and 4B models (Gemma Team et al., 2025) on in-distribution and out-of-distribution evaluations. We compare intrinsic dimensionality against alternative metrics based on length of trajectories and likelihood under the student model, finding that intrinsic dimensionality provides substantially stronger predictive power. These findings provide a principled, quantitative predictor of the extent to which reasoning strategies improve generalization, and offer potential guidance for data annotation, model alignment, and training optimization.

**Conflict of Interest Disclosure.** Authors AP, MJ, KL, and PS were employed by Google DeepMind, which leads the development of the Gemma models used in this paper.

## 2. Intrinsic Dimensionality of Reasoning

### 2.1. Background on Intrinsic Dimension

The concept of intrinsic dimensionality formalizes the observation that many tasks can be learned in lower-dimensional subspaces than the full parameter space of overparameterized neural networks. Following Li et al. (2018); Aghajanyan et al. (2021), we can express the model's parameters $\theta \in \mathbb{R}^D$ as: $\theta = \theta_0 + P(\theta_d)$, where $\theta_0$ represents the pretrained model parameters, $D$ is the total number of parameters in the model, $\theta_d \in \mathbb{R}^d$ is a lower-dimensional parameter vector with $d \leq D$, and $P : \mathbb{R}^d \to \mathbb{R}^D$ is a projection operator. By training only in this $d$-dimensional subspace, we can identify the minimum dimensions $d$ required to achieve a target performance, which defines the intrinsic dimension of the task under a given model.

### 2.2. Lower-Dimension Projection for LLMs

While the original formulation uses random projections applied globally to all parameters, optimizing in such randomly projected spaces can be challenging and sub-optimal for larger models (Aghajanyan et al., 2021; Hu et al., 2022). Instead, we adopt the Low-Rank Adaptation framework (LoRA; Hu et al., 2022), which was itself motivated by the intrinsic dimension findings of Li et al. (2018) and Aghajanyan et al. (2021) and has proven effective for fine-tuning LLMs. LoRA targets specific weight matrices in the transformer architecture and constrains their updates to low-rank subspaces. For a pretrained weight matrix $W_0 \in \mathbb{R}^{m \times n}$, LoRA represents the weight update as:

$$W = W_0 + BA$$

where $B \in \mathbb{R}^{m \times r}$ and $A \in \mathbb{R}^{r \times n}$ are learned low-rank matrices with rank $r \leq \min(m, n)$. During training, $W_0$ remains frozen while $B$ and $A$ are optimized. LoRA can be applied to different subsets of weight matrices, including attention modules ($W_q, W_k, W_v, W_o$), MLP layers, or all transformer layers. The total number of trainable parameters, which we denote as $\mathrm{params}(.)$, is determined by the rank and the number of weight matrices:

$$\mathrm{params}(r, L_{LoRA}) = 2 \times L_{LoRA} \times d_{model} \times r$$

where $L_{LoRA}$ is the number of weight matrices LoRA is applied to, $d_{model}$ is the model's hidden dimension, and $r$ is the LoRA rank. This formulation aligns with intrinsic dimensionality by using LoRA as a structured low-dimensional projection, constraining trainable capacity in a manner that is both architecturally informed and empirically effective for LLM fine-tuning.

### 2.3. Measuring Intrinsic Dimension of Reasoning

We measure intrinsic dimensionality as the *minimum number of trainable parameters* required to reach a specified performance threshold. Formally, for a task with performance metric $\mathcal{A}$, the intrinsic dimension $d_{int}$ is:

$$d_{int} = \min\{d : \mathcal{A}(d) \geq \tau\}$$

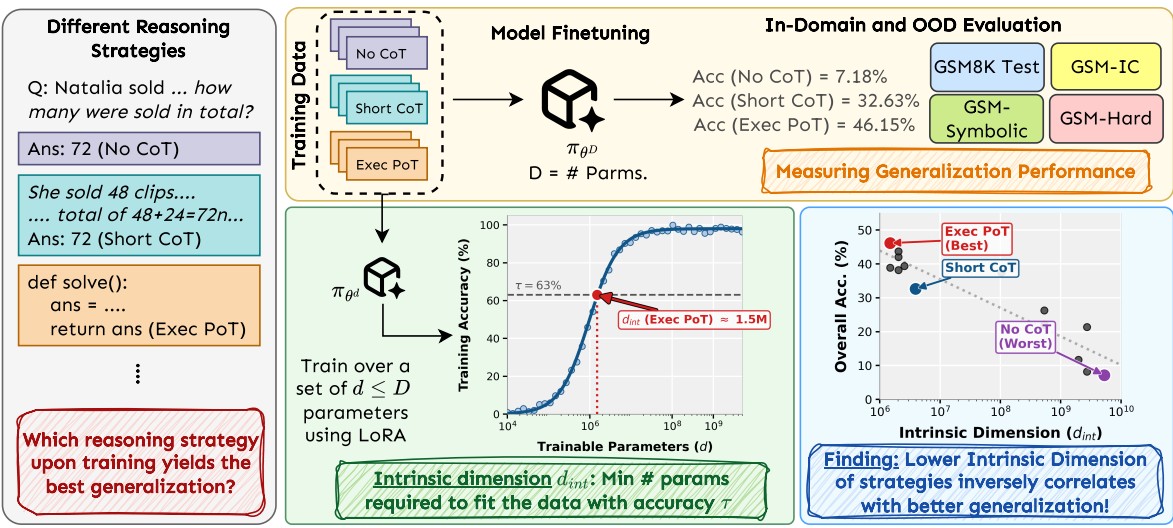

*Figure 1.* **Overview. Top:** We compare intrinsic dimensionality across reasoning strategies and evaluate how well it predicts generalization performance (c.f. Section 3). **Middle (Green):** To compute the intrinsic dimension of a reasoning strategy, we sweep varying LoRA parameter counts $d$ and identify the minimum $d_{int}$ at which training accuracy crosses a common threshold $\tau$ (described in Section 2.3). This Pareto frontier for a single strategy is shown across all strategies simultaneously in Figure 2. **Right:** We find a *strong inverse correlation* between intrinsic dimensionality and generalization performance (Section 4).

where $d = \mathrm{params}(r, L_{LoRA})$ is the total number of trainable parameters, $\mathcal{A}(d)$ is the *training accuracy* achieved with this configuration, and $\tau$ is the performance threshold. To compute $d_{int}$, we conduct a sweep of $k$ LoRA configurations where we vary both the rank $r$ and which weight matrices receive LoRA adaptations (controlled by $L_{LoRA}$). The configurations are chosen such that the resulting parameter counts $\mathrm{params}(r, L_{LoRA})$ are uniformly distributed in log scale. The lower bound of our sweep applies LoRA of rank $r = 1$ to only the query and value projection matrices in the attention modules ($W_q$ and $W_v$). The upper bound applies full-rank LoRA ($r = d_{model}$) to all layers (both attention and MLP modules), for additional details we refer readers to Appendix A.1. For each configuration, we train the model and record the training accuracy at the completion of training, then identify the minimum parameter count at which training accuracy exceeds the threshold $\tau$.

**Choice of Performance Threshold.** Prior work on intrinsic dimension measurement sets $\tau$ as a percentage (e.g., 90%) of the validation performance achieved with full finetuning for a fixed model and training data (Li et al., 2018; Aghajanyan et al., 2021). However, our work differs crucially: we fix the model but implicitly vary the data by changing the reasoning strategy (CoTs) in the output space, resulting in $T$ different training datasets, one for each CoT strategy applied to the same set of problems. Different reasoning strategies may differ in the maximum attainable performance, making percentage-based thresholds incomparable. Rather than focusing on the absolute value of intrinsic

dimensionality, we argue that the *relative ordering of intrinsic dimensions across different reasoning strategies* is what matters for understanding their effectiveness. Therefore, we use a common threshold $\tau$ across all strategies to ensure fair comparison. We set $\tau$ based on either: (1) a fixed validation accuracy threshold representing strong reasoning performance, or (2) the full-capacity training accuracy after one epoch (which avoids overfitting contamination), allowing us to compute the intrinsic dimension and thereby, identifying *effective reasoning strategies entirely from training curves alone*. Unless stated otherwise, we set $\tau$ based on the latter, computed as the *maximum* training accuracy *achieved by any strategy* at full capacity after one epoch of training. We evaluate each strategy's sweep against this same threshold, identifying the minimum parameter count needed to reach $\tau$, as illustrated in Figure 1; thus measuring how much capacity each reasoning approach requires to achieve the same level of capability. In Section 4.3, we justify this choice and demonstrate that conclusions drawn from intrinsic dimensions are generally robust to the exact choice of threshold, holding across a wide range of thresholds.

## 3. Experimental Setup

**Datasets.** We use the training split of the well-studied GSM8K dataset (Cobbe et al., 2021) comprising grade-school level math word problems. To measure models' abilities at solving word problems in general, we evaluate the trained models on the (i) in-domain test split of GSM8K, as well as several stress test sets that measure out-of-domain

generalization (ii) GSM-Symbolic (Mirzadeh et al., 2025), (iii) GSM-IC (Shi et al., 2023), and (iv) GSM-Hard (Gao et al., 2023). Mirzadeh et al. (2025) propose GSM-Symbolic to test robustness of models on a diverse set of questions sampled from symbolic perturbations to the question's phrasing and varying difficulty via three different splits. Shi et al. (2023) find that performance of models on math word problems is diminished in the presence of irrelevant sentences in the question which have no bearing on the solution. Finally, Gao et al. (2023) measure the numerical robustness and the ability to solve word problems involving more complex arithmetic. We use the test split of the GSM8K dataset to measure the in-distribution (ID) performance, and report the geometric mean of the 5 stress test sets as the out-of-distribution (OOD) performance. The overall performance is computed as the geometric mean across all the 6 test splits. We enumerate the size of various test splits in Appendix B.1.

**Reasoning Strategies.** We evaluate intrinsic dimensionality across a diverse set of reasoning strategies that vary in length, structure, and generation method. Our simplest baselines are **No CoT**, which outputs a direct answer without intermediate reasoning (Sprague et al., 2025), and **No CoT with extra tokens**, which appends filler text to isolate the effect of inference-time computation from reasoning quality. Using Gemma-3 27B, we generate three natural-language CoT variants: **Very Short CoT**, prompted for concise, equation-style reasoning (Nye et al., 2022); **Short CoT**, restricted to brief (1–2 sentence) explanations; and **Gemma 27B CoT**, which allows unconstrained reasoning. In contrast, **Gemini CoT** is produced by a stronger teacher model known for longer solutions (Gemini 2.5 Flash; Comanici et al., 2025). To study robustness to irrelevant information, we include **Short CoT with $n$ Distractors** ($n \in 2, 4, 8$), where unrelated steps sampled from other problems are inserted before reaching the correct answer (Li et al., 2025). We additionally evaluate code- and structure-based approaches, including **Executed PoT** with actual program execution (Gao et al., 2023; Chen et al., 2023), **Simulated PoT** relying on internal code simulation (Sprague et al., 2025), and **Plan and Solve** following a decomposition framework (Wang et al., 2023b). Finally, we evaluate **Critical CoT** (Zhou et al., 2024), a reasoning structure associated with critical-thinking strategies, and **High Review Ratio CoT** (Feng et al., 2025), with higher occurrences of revision tokens for longer and verification-based reasoning.

We provide examples of each strategy in Appendix E. All strategies except Gemini CoT are generated by prompting instruction-tuned Gemma-3 27B for each reasoning style and filtering out generations with incorrect final answers (see Appendix B.1). Together, these strategies span a broad range of lengths and structural properties, enabling us to test whether intrinsic dimensionality explains reasoning effectiveness beyond metrics such as trajectory length. To measure generalization performance of each strategy, we curate a training dataset for each reasoning strategy based on the train split of GSM8K (see details in Appendix B.1) and finetune a full-capacity model to generate a rationale (using the above strategies) followed by the final answer.

**Baseline Metrics.** In addition to intrinsic dimensionality (unless mentioned otherwise, computed with a threshold of 90% of maximum training accuracy attained by any strategy after the first epoch), we compare against several baseline metrics that do not require test-time evaluation to assess their ability to predict generalization performance:

1. **Trajectory Length**: Since longer responses increase inference-time computation and often correlate with better reasoning (e.g., via backtracking or verification) (Snell et al., 2024; Guo et al., 2025; Marjanović et al., 2025), we measure the average token length of CoTs to test if length alone predicts effectiveness.

2. **Token Perplexity**: Recent work (Yue et al., 2026; Karan & Du, 2026; Zhang et al., 2025) suggests that proximity or overlap between the pretrained distribution and fine-tuning data affects learning effectiveness, i.e., models learn more effectively from reasoning chains that are in-distribution to the base model (Agarwal et al., 2024; Yue et al., 2026). We compute the average token-level perplexity of CoT training data for each strategy relative to the pretrained student model.[1]

3. **Sequence KL Divergence**: To account for full-sequence probability rather than per-token averages, we estimate the KL divergence between the empirical data distribution $\hat{\pi}$ (uniform over the $N$ training samples) and the student model's distribution $\pi_\theta$. Emprically, this is computed as the average sequence-level negative log-likelihood: $\frac{1}{N} \sum_{i=1}^{N} -\log \pi_\theta(y^{(i)}|x^{(i)})$. Unlike token perplexity, this metric is not normalized by sequence length, making the two metrics complementary measures of distribution alignment.

**LoRA Training Hyperparameters.** We fine-tune Gemma-3 base models (Gemma Team et al., 2025) of two sizes: 1B and 4B parameters, using consistent hyperparameters across all reasoning strategies. We train the 1B model for 8,000 steps (learning rate $10^{-3}$) and the 4B model for 6,000 steps ($10^{-4}$); these parameters were set based on preliminary full-capacity training runs to ensure training accuracy fully plateaus. We employ the AdamW optimizer (Loshchilov & Hutter, 2019) with a training batch size of 8, evaluating on the validation set to select checkpoints for ID and OOD performance reporting. We sweep across $k = 20$ (1B) and $k = 30$ (4B) LoRA configurations, with parameter counts distributed uniformly in log space from

---

[1]Note that our setting uses the likelihood of the token under the teacher model to be 1, and could be relaxed to use the teacher probabilities if available. We leave this comparison to future work.

*Table 1.* Performance of Gemma-3 4B across reasoning strategies. **ID**: Accuracy on GSM8K Test. **OOD**: Geometric mean of 5 stress test sets (GSM-Symbolic, GSM-IC, GSM-Hard). **Overall**: Geometric mean across all 6 splits (refer to Table 8 for a breakdown of accuracy across splits). Spearman correlations are computed between each metric and overall accuracy; for metrics marked with ↓, +ve correlations indicate that lower values successfully predict higher accuracy. Note: $^\dagger$ denotes statistical significance ($p < 0.01$).

| CoT Strategy | ID | OOD | Overall | Intrinsic Dim. (M) ↓ | KL Div ↓ | Token PPL ↓ | Length ↑ |
|---|---|---|---|---|---|---|---|
| *Baseline Strategies* | | | | | | | |
| No CoT | 14.94 | 6.20 | 7.18 | 5246.16 | 45.46 | 91.34 | 9.31 |
| No CoT with extra tokens | 16.45 | 7.15 | 8.22 | 2729.64 | 123.00 | 59.06 | 23.31 |
| *Short CoT Variants* | | | | | | | |
| Very Short CoT | 44.58 | 19.22 | 22.11 | 532.81 | 89.69 | 6.84 | 42.63 |
| Short CoT | 58.98 | 28.99 | 32.63 | 3.92 | 116.31 | 2.73 | 93.53 |
| Short CoT with 2 Distractors | 50.11 | 23.09 | 26.27 | 532.81 | 494.42 | 3.43 | 289.97 |
| Short CoT with 4 Distractors | 41.32 | 18.74 | 21.38 | 2729.64 | 775.23 | 3.12 | 485.11 |
| Short CoT with 8 Distractors | 22.97 | 10.21 | 11.69 | 1968.84 | 1315.04 | 2.86 | 878.82 |
| *Default CoTs Sampled from Teacher Model* | | | | | | | |
| Gemma 27B CoT | 67.48 | 38.24 | 42.04 | 2.05 | 162.42 | 1.84 | 221.95 |
| Gemini CoT | 66.72 | 35.46 | 39.40 | 2.55 | 571.86 | 1.90 | 650.72 |
| *Specific Reasoning Strategies from Prior Work* | | | | | | | |
| Executed PoT | 62.77 | **43.40** | **46.15** | 1.49 | 131.30 | 1.77 | 188.79 |
| Simulated PoT | 64.75 | 35.13 | 38.90 | 1.49 | 257.91 | 1.67 | 388.11 |
| Plan Solve | 64.75 | 34.33 | 38.16 | 2.05 | 250.16 | 1.77 | 333.04 |
| Critical CoT | 63.84 | 33.74 | 37.52 | 104.12 | 924.26 | 3.07 | 591.39 |
| High Review Ratio CoT | **67.63** | 40.10 | 43.75 | 2.05 | 727.69 | 2.63 | 547.13 |
| **Spearman Rank Correlation** | - | - | - | $0.93^\dagger$ | -0.17 | $0.82^\dagger$ | 0.31 |

minimum (rank 1 applied to query and value projection matrices only) to maximum (full rank applied to all attention and MLP layers). The best configuration for each parameter target is selected by minimizing the absolute difference between the target and actual trainable parameter count (c.f. Appendix A.1).

## 4. Main Results and Analysis

### 4.1. Intrinsic Dimension with Gemma-3 4B

**Setup.** The goal of our study is to measure the extent to which intrinsic dimensionality and other baselines are predictive of the generalization performance of Gemma-3 4B under different reasoning strategies. To this end, we compute the Spearman rank correlation between the average performance (including in-distribution as well as out-of-distribution datasets) and each metric as reported in Table 1. For metrics where smaller values theoretically indicate better learnability (Intrinsic Dimension, KL Divergence, Token Perplexity), we compute correlation between increasing metric values and decreasing average accuracy, such that positive correlations indicate successful prediction in the theoretically expected direction; for instance, a correlation of 0.93 for Intrinsic Dimension means strategies with lower intrinsic dimensionality achieve higher accuracy. For Length, we report standard correlation with average accuracy.

**Intrinsic Dimension Strongly Predicts Reasoning Effectiveness.** Table 1 presents our main results across 14 reasoning strategies on the Gemma 4B model. Intrinsic di-

mensionality exhibits the strongest correlation with generalization performance, achieving a Spearman rank correlation of 0.93 with average accuracy – substantially higher than all baseline metrics. This demonstrates that effective reasoning chains require significantly fewer parameters to learn, supporting our hypothesis that such chains help models learn more compressible task representations.

**Length and KL Divergence Show Poor Predictive Power.** In contrast, length shows weak correlation (0.31) with reasoning effectiveness, which is unsurprising given conflicting evidence in prior work (Feng et al., 2025; Snell et al., 2024; Wu et al., 2026) showing that depending on the task, there may be an optimal reasoning length beyond which performance worsens. Similarly, KL divergence exhibits weak or negative correlation (-0.17). This can be explained by noting that KL divergence directly measures the cost in bits to encode the training data (CoT + answer) using the model (Blier & Ollivier, 2018). While this metric is effective for comparing different models on the same task and output space, it is prohibitive for comparing effectiveness across reasoning strategies: irrespective of how easy the trajectory is to encode in terms of likelihood, it's length unduly influences the divergence, preferring shorter lengths and leading to no meaningful correlation.

**Connection between Token Perplexity and Intrinsic Dimensionality.** In Table 1, token perplexity achieves a correlation of 0.82, though still lower than intrinsic dimensionality's 0.93. One way of understanding this relationship is that the two metrics are potentially interrelated: reasoning

*Table 2.* Performance of Gemma-3 1B across reasoning strategies. **ID**: Accuracy on GSM8K Test. **OOD**: Geometric mean of 5 stress test sets (GSM-Symbolic, GSM-IC, GSM-Hard). **Overall**: Geometric mean across all 6 splits (refer to Table 9 for a breakdown of accuracy across splits). Spearman correlations are computed between each metric and overall accuracy; for metrics marked with ↓, +ve correlations indicate that lower values successfully predict higher accuracy. Note: [†] denotes statistical significance ($p < 0.05$).

| CoT Strategy | ID | OOD | Overall | Intrinsic Dim. (M) ↓ | KL Div ↓ | Token PPL ↓ | Length ↑ |
|---|---|---|---|---|---|---|---|
| *Baseline Strategies* | | | | | | | |
| No CoT | 3.56 | 2.00 | 1.84 | 119.38 | 58.74 | 236.16 | 9.31 |
| No CoT with extra tokens | 5.31 | 2.00 | 2.38 | 83.03 | 148.07 | 121.88 | 23.31 |
| *Short CoT Variants* | | | | | | | |
| Very Short CoT | 8.95 | 4.00 | 4.39 | 31.45 | 136.15 | 15.78 | 42.63 |
| Short CoT | 18.04 | 7.00 | 8.68 | 1.03 | 191.41 | 4.89 | 93.53 |
| Short CoT with 2 Distractors | 10.46 | 5.00 | 5.44 | 7.34 | 662.68 | 5.16 | 289.97 |
| Short CoT with 4 Distractors | 4.78 | 3.00 | 2.91 | 31.45 | 1028.70 | 4.49 | 485.11 |
| Short CoT with 8 Distractors | 2.43 | 1.00 | 1.57 | 134.93 | 1740.59 | 4.01 | 878.82 |
| *Default CoTs Sampled from Teacher Model* | | | | | | | |
| Gemma 27B CoT | 20.40 | 7.00 | 8.27 | 7.34 | 286.24 | 2.82 | 221.95 |
| Gemini CoT | 20.55 | 8.00 | 9.73 | 31.45 | 854.37 | 2.62 | 650.72 |
| *Specific Reasoning Strategies from Prior Work* | | | | | | | |
| Executed PoT | 20.24 | **11.00** | **11.76** | 1.03 | 247.16 | 2.83 | 188.79 |
| Simulated PoT | 20.85 | 8.00 | 8.98 | 7.34 | 431.72 | 2.31 | 388.11 |
| Plan Solve | 21.53 | 10.00 | 11.24 | 7.34 | 432.53 | 2.63 | 333.04 |
| Critical CoT | 17.51 | 8.00 | 9.11 | 31.45 | 1318.52 | 4.91 | 591.39 |
| High Review Ratio CoT | **22.60** | 9.00 | 10.57 | 7.34 | 1042.57 | 3.95 | 547.13 |
| **Spearman Rank Correlation** | - | - | - | 0.75[†] | -0.18 | 0.63[†] | 0.24 |

chains that exhibit high likelihood and low surprisal under the base model are likely also readily compressible with fewer parameters that need to be altered. This is consistent with the findings in Yue et al. (2026), which shows that even after reinforcement-learning, the reasoning chains of the trained model still exhibit decreased perplexity under the base model, suggesting that effective reasoning remains grounded in the base model's distribution.

### 4.2. Intrinsic Dimension with Gemma-3 1B

Table 2 presents results for the Gemma 1B model across the same 14 reasoning strategies. Despite the 1B model achieving substantially lower absolute performance than the 4B model, intrinsic dimensionality maintains a strong correlation of 0.75 with generalization performance. This demonstrates that the predictive power of intrinsic dimensionality holds even when the performance ceiling is significantly lower, suggesting the metric captures fundamental properties of reasoning strategy effectiveness that are independent of model scale. Token perplexity remains the second-best predictor with a correlation of 0.63, while length (0.24) and KL divergence (-0.18) continue to show poor predictive power. This is consistent with our findings on Gemma-3 4B that intrinsic dimensionality outperforms other baselines.

### 4.3. Robustness to Threshold Selection

**Setup.** Recall that in Sections 2.3 and 3, we compute intrinsic dimensionality by identifying the minimum param-

*Table 3.* Robustness to threshold selection for computing intrinsic dimension using epoch 1 training accuracy at 70%, 80%, and 90% of the maximum achieved by any strategy, and 90% of validation accuracy by any strategy.

| Threshold Selection | Correl: 1B | Correl: 4B |
|---|---|---|
| 90% of Epoch 1 Train Acc | 0.75 | 0.93 |
| 80% of Epoch 1 Train Acc | 0.72 | 0.94 |
| 70% of Epoch 1 Train Acc | 0.73 | 0.93 |
| 90% of Val Acc | 0.73 | 0.94 |

eter count needed to reach a common threshold $\tau$ across all reasoning strategies. For the main results presented in Tables 1 and 2, we use the maximum full-capacity training accuracy after one epoch across reasoning strategies at 90% as our threshold, following prior work in computing intrinsic dimension for other domains (Li et al., 2018; Aghajanyan et al., 2021). However, a critical question remains: are our findings dependent on a particular "golden" or ad hoc choice of threshold? To address this concern, we evaluate the robustness of intrinsic dimensionality's predictive power across different threshold selection methods and threshold levels. Even if individual intrinsic dimension measurements become noisier with different thresholds, we expect the relative ordering of reasoning strategies, and thus, the correlation with performance, to remain stable if intrinsic dimensionality reliably measures the effectiveness of different reasoning strategies.

**Strong Correlations Persist Across Thresholds.** Table 3 demonstrates that our findings are remarkably robust to

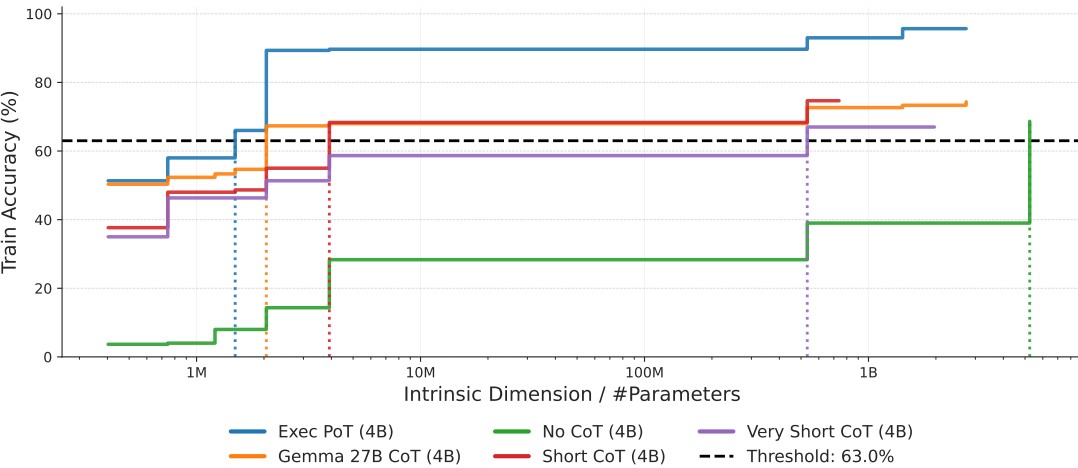

*Figure 2.* Visualization of intrinsic dimension computation for Gemma-3 4B showing select reasoning strategies. We plot the Pareto frontier of monotonic training accuracy versus trainable parameters (log scale). The dashed line indicates the threshold ($\tau = 63.0\%$); intrinsic dimension is the parameter count where each curve first crosses this threshold (vertical dotted lines). Strategies crossing earlier have lower intrinsic dimensionality and tend to yield higher overall performance (cf. Table 1).

threshold selection. Across both the 1B and 4B models, intrinsic dimensionality maintains strong correlations (ranging from 0.72 to 0.94) whether we use epoch 1 training accuracy at 70%, 80%, or 90% thresholds, or validation accuracy at 90%. The consistency across threshold levels confirms that the predictive power of intrinsic dimensionality is not an artifact of hyperparameter tuning. Notably, using epoch 1 training accuracy offers a practical advantage: it enables computing *intrinsic dimensionality entirely from training curves* without requiring validation set evaluation, making the metric more accessible (e.g., when most of the data is used for training, or out-of-distribution testing is not feasible) and reducing computational overhead while maintaining strong predictive performance. Additionally, it avoids contamination in the threshold selection from overfitting or memorization that occurs at later training stages.

### 4.4. Additional Results and Analysis

To compare different reasoning strategies fairly, we enforce a common absolute accuracy threshold across all strategies (c.f. Section 2.3). In Figures 2 and 3 this corresponds to $\tau = 63.0\%$ for 4B and $\tau = 24.3\%$ for 1B models; representing a realistic gauge of generalization capability, and explains the gap between the thresholds for 1B and 4B models. We derive $\tau$ from the maximum training accuracy achieved by any strategy after the first epoch. This choice is critical: while final training accuracy often conflates generalizable learning with rote memorization (overfitting), performance after the first epoch effectively isolates the *learnability* of the reasoning structure.

**Executed PoT Achieves Lowest Intrinsic Dimensionality and Best OOD Performance.** Figures 2 and 3 visualize the intrinsic dimensionality computation by plotting the

Pareto frontier of training accuracy as a function of parameter count for select reasoning strategies. Examining these curves and the results in Tables 1 and 2, we observe that Executed PoT consistently achieves both the lowest intrinsic dimensionality and the strongest out-of-distribution performance across both model sizes. For the 4B model (Figure 2), Exec PoT crosses the 63.0% threshold at only 1.49M parameters – substantially lower than other strategies shown: Short CoT (3.92M), Gemma 27B CoT (2.05M), Very Short CoT (532.81M), and even the No CoT baseline (at full capacity or rank), while achieving the highest OOD accuracy of 43.40% (Table 1). This finding aligns with Sprague et al. (2025), who demonstrate that python code + interpreter solutions often outperform CoT in zero-shot settings across multiple models. The combination of low intrinsic dimensionality and strong generalization suggests that code-based reasoning with execution provides a particularly compressible and robust representation of our task.

**Larger Models Compress Effective Reasoning Strategies More Efficiently.** Comparing intrinsic dimensionality across model sizes in Tables 1 and 2 reveals an interesting pattern: for effective reasoning strategies, the 4B model exhibits lower intrinsic dimensionality than the 1B model despite having a larger parameter space. For instance, comparing the training accuracy curves in Figures 2 and 3, Executed PoT requires 1.49M parameters for the 4B model versus 1.03M for the 1B model, but critically, the 4B model achieves this at a much higher absolute performance level (maximum accuracy of 46.15% vs 11.76%, and threshold of 63.0% vs 24.3%). Similarly, Gemma 27B CoT and other effective strategies show comparable or lower intrinsic dimensionality on the 4B model relative to their task complexity. This demonstrates that larger models, despite being more overparameterized, are better compressors of effective rea-

*Table 4.* Performance on Reasoning Gym tasks for Gemma-3 4B across algorithmic and cognitive reasoning categories. **ID:** Geometric mean of accuracies across three tasks, **OOD:** geometric mean over out-of-distribution splits with hard configuration and held-out tasks **Overall:** denotes the geometric mean across all evaluation splits. Spearman correlations are computed between each metric and overall accuracy across the four strategies. For metrics marked with ↓, +ve correlations indicate lower values predict higher accuracy.

| | CoT Strategy | ID | OOD | Overall | Intrinsic Dim. (M) ↓ | KL Div ↓ | Token PPL ↓ | Length ↑ |
|---|---|---|---|---|---|---|---|---|
| *Algorithmic* | No CoT | 22.53 | 4.49 | 7.69 | 532.81 | 71.26 | 7.33 | 28.63 |
| | Short CoT | 51.60 | 11.40 | 18.86 | 3.92 | 105.75 | 1.78 | 140.88 |
| | Gemma 27B CoT | 57.71 | 11.68 | 19.90 | 1.49 | 39.44 | 1.20 | 191.79 |
| | Executed PoT | **96.67** | **63.71** | **73.21** | 0.74 | 52.43 | 1.28 | 176.52 |
| | **Spearman Rank Correl.** | - | - | - | 1.00 | 0.60 | 0.80 | 0.80 |
| *Cognitive* | No CoT | **61.93** | **44.68** | **49.82** | 0.40 | 51.68 | 15.43 | 6.99 |
| | Short CoT | 51.85 | 37.99 | 42.15 | 2.05 | 145.79 | 5.75 | 62.12 |
| | Gemma 27B CoT | 44.87 | 36.06 | 38.78 | 3.92 | 113.79 | 1.60 | 182.89 |
| | Executed PoT | 51.14 | 35.04 | 39.74 | 2.55 | 112.41 | 1.61 | 178.67 |
| | **Spearman Rank Correl.** | - | - | - | 1.00 | 0.40 | -1.00 | -1.00 |

soning tasks – consistent with findings in existing intrinsic dimensionality work (Aghajanyan et al., 2021) showing that larger networks learn more efficient representations.

**Ineffective Reasoning Strategies Reveal Higher Intrinsic Dimensionality in Larger Models.** Interestingly, the pattern reverses for less effective reasoning strategies. As shown in Tables 1 and 2, strategies like No CoT, No CoT with extra tokens, and Very Short CoT with multiple distractors exhibit considerably higher intrinsic dimensionality on the 4B model compared to the 1B model. For example, Very Short CoT requires over 500M parameters for Gemma-3 4B model but only 31M on the 1B model, as visible in the delayed threshold crossing in Figure 2 vs. Figure 3. This suggests that when reasoning chains provide little structure or contain substantial noise (as with distractors), larger models require disproportionately more capacity to fit these less compressible patterns, while smaller models may more readily resort to simpler memorization strategies that require lesser deviation from the base model.

*Table 5.* Effect of training data correctness on intrinsic dimensionality and generalization (Gemma-3 4B). Mixed sets consist of 50% correct and 50% incorrect reasoning chains.

| CoT Strategy | % Correct | Overall Acc. (↑) | $d_{int}$ (M) (↓) |
|---|---|---|---|
| Short CoT | 100% | 32.63 | 3.92 |
| Short CoT | 50% | 20.61 | 2729.64 |
| Gemma 27B CoT | 100% | 42.04 | 2.05 |
| Gemma 27B CoT | 50% | 29.24 | 257.18 |

**Effect of Training Data Correctness on Intrinsic Dimensionality.** Our framework measures the intrinsic dimensionality of a *reasoning strategy*, but a natural question is whether this is sensitive to the correctness of individual training instances. To investigate, we construct mixed training sets for Short CoT and Gemma 27B CoT, where 50% of instances contain incorrect reasoning chains (i.e., chains leading to a wrong final answer), with the remaining 50%

correct. As shown in Table 5, mixing in incorrect reasoning chains substantially increases intrinsic dimensionality and leads to a corresponding drop in generalization performance for both strategies. For Short CoT, intrinsic dimensionality increases by over a factor of $500\times$ with a 12.0 point drop in accuracy; whereas, for Gemma 27B CoT, the increase is $\sim$125$\times$ with a 12.8 point drop. This is consistent with our interpretation: incorrect reasoning chains do not provide a consistent, compressible input-output mapping – the model cannot exploit a coherent low-dimensional structure to fit the task, and instead requires significantly more parameters to accommodate the contradictory signal.

### 4.5. Generalization to Diverse Reasoning Domains

**Setup.** To assess whether our findings generalize beyond mathematical reasoning, we evaluate intrinsic dimensionality on tasks from Reasoning Gym (Stojanovski et al., 2025), selecting two categories testing qualitatively different reasoning skills: *algorithmic reasoning* (base_conversion, leg_counting, rotate_matrix) and *cognitive reasoning* (color_cube_rotation, number_sequence, rectangle_count). For each category, we report accuracy on (i) an in-distribution (**ID**) test set using easy task configurations, (ii) an out-of-distribution (**OOD**) test set using harder configurations of the same tasks as well as held-out task types (algorithmic: letter_counting, count_primes; cognitive: needle_haystack, modulo_grid), both aggregated via geometric means across tasks and splits. We evaluate a representative subset of strategies on Gemma-3 4B. We detail the dataset configurations in Appendix C.

**Intrinsic Dimensionality Remains the Strongest Predictor Across Domains.** Table 4 presents results across both reasoning categories. For *algorithmic* tasks, Executed PoT achieves the lowest intrinsic dimensionality (0.74M) and the best overall performance (73.21), while No CoT requires the most capacity (532.81M) and performs worst – closely

mirroring our GSM8K findings. For *cognitive* tasks, the pattern is distinctive: No CoT achieves the lowest intrinsic dimensionality (0.40M) and best overall performance (49.82). This is in line with Sprague et al. (2025) who show CoT provides limited benefit on knowledge-intensive tasks. Across both categories, KL divergence and length show weaker and less consistent correlations, further supporting intrinsic dimensionality as the most reliable predictor.

## 5. Related Work

**Evaluation and Analysis of Reasoning.** Recent work has aimed to disentangle the factors driving the efficacy of Chain-of-Thought (CoT) reasoning, revealing that performance gains are highly task-dependent – often providing limited benefits for knowledge-intensive tasks (Sprague et al., 2025) – and are driven more by the structural coherence of the reasoning template than by local numerical precision (Wang et al., 2023a; Li et al., 2025). However, many of these explanations are not quantifiable measures, and for those that are quantifiable, our meta-evaluation in Section 4 shows that intrinsic dimensionality offers stronger correlation and is more predictive of downstream gains in reasoning performance of the target model. A parallel line of work has developed various proxies for selecting high-quality reasoning chains, ranging from simple heuristics like ensemble-based agreement (Wang et al., 2023c), to those based on reasoning length and review tokens (Feng et al., 2025), high-entropy "forking" tokens (Wang et al., 2025), and finer-grained linguistic or information-theoretic scores (Golovneva et al., 2023; Prasad et al., 2023). However, these metrics primarily serve as proxies for instance-level *correctness* rather than explaining what makes a reasoning strategy effectively *learnable*. In contrast, we show that intrinsic dimensionality – grounded in concept of minimum description length – provides a principled, quantitative metric, where lower intrinsic dimensionality consistently predicts stronger generalization.

**Intrinsic Dimension of Neural Networks.** In the context of deep learning, Li et al. (2018) and Aghajanyan et al. (2021) proposed the concept of *intrinsic dimensionality*, quantifying the minimum degrees of freedom required to optimize a network for a specific objective. This line of research demonstrated that large, pretrained models possess remarkably low intrinsic dimensions, which directly motivated parameter-efficient fine-tuning methods such as LoRA (Hu et al., 2022). In this work, we invert this experimental setup: rather than varying the model to study pre-training quality, we vary the LoRA rank to empirically measure the intrinsic dimension of different reasoning strategies. We distinguish our method from measuring the intrinsic dimension of the data manifold itself (Levina & Bickel, 2004; Tenenbaum et al., 2000). While the latter estimates

the complexity of the training data and which scales with dataset size, we estimate the complexity of the *learning objective* itself, involving both the model and the reasoning task. We posit that effective reasoning strategies simplify the underlying rule connecting inputs to answers, enabling the model to fit the task within a lower-dimensional subspace despite the increased length of the output. This notion of intrinsic dimensionality is related to the Minimum Description Length (MDL) principle (Rissanen, 1978; Grünwald, 2007; Hinton & Van Camp, 1993), which frames learning as data compression. If accuracy is held constant, then the MDL principle suggestions that the best model is the one with the shortest description length.

## 6. Discussion and Conclusion

We establish that the generalization performance of a model trained on a given set of reasoning chains is correlated with the degree to which the reasoning chains reduce the intrinsic dimensionality of the given task. This offers a new perspective on why chain-of-thought reasoning can improve generalization, grounded in information theory and the minimum description length (MDL) principle: *effective reasoning chains reduce the conditional complexity of learning a new task*. This measure of intrinsic dimensionality correlates better with generalization performance than alternatives such as perplexity and length. Notably, while measures such as perplexity are computed over individual trajectories and then aggregated, intrinsic dimensionality potentially captures coherence across the complete set of trajectories, and future work could investigate the importance of this aspect further. While we focus on fine-tuning models given reasoning trajectories, future work could explore other post-training settings (e.g., Zelikman et al., 2022; Agarwal et al., 2024).

From a practical perspective, our method to estimate intrinsic dimensionality requires fine-tuning adapters of various sizes, and therefore this measure would be computationally expensive to optimize for directly. Future work could potentially build on these insights to explore more computationally tractable alternatives for identifying effective reasoning chains that enable greater generalization. From a theoretical standpoint, any layer-wise projection or PEFT method provides an upper bound on intrinsic dimensionality and may serve as an acceptable proxy (Li et al., 2018; Aghajanyan et al., 2021), although the bound may be loose relative to the underlying Kolmogorov complexity of the data for certain tasks (Shaw et al., 2026). Prior work has also compared the compression ratios of LoRA (Hu et al., 2022) and subspace training methods, yielding even tighter bounds (Lotfi et al., 2024). Future work could empirically verify whether alternative PEFT methods yield consistent relative orderings of reasoning strategies.

## Acknowledgments

We sincerely thank Jacob Eisenstein, and Kristina Toutanova for their valuable feedback on early drafts of this work. Part of this work was done during an internship at Google DeepMind. This work was partially supported by NSF-AI Engage Institute DRL2112635, NSF-CAREER Award 1846185, DARPA ECOLE Program No. HR00112390060, and an Apple PhD Fellowship. The views contained in this article are of the authors and not of the funding agency.

## Impact Statement

This paper presents work whose goal is to advance the understanding of reasoning in Large Language Models through the lens of intrinsic dimensionality. Our findings are primarily analytical and interpretive in nature, and we do not believe our method introduces risks beyond those inherent in existing LLM reasoning research. That said, intrinsic dimensionality is a statistical measure that may reflect artifacts of specific model families or datasets, and should be interpreted with care. We hope this work provides the research community with complementary tools to better characterize, evaluate, and ultimately improve the reliability of reasoning in language models.

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

# A. Additional Details on Computing Intrinsic Dimension

## A.1. LoRA Sweeps for Computing Intrinsic Dimensions

To compute intrinsic dimensionality, we conduct a sweep over LoRA configurations with parameter counts uniformly distributed in log scale. The key challenge is that different combinations of rank $r$ and target modules $L_{LoRA}$ can yield similar parameter counts but potentially different performance. To address this, we employ a greedy search procedure that selects the configuration minimizing the absolute error between the target and actual parameter count for each point in our sweep. We define four groups of target modules that represent increasing levels of model adaptation:

- **attention_q_v**: Query and value projections only ($W_q$, $W_v$)
- **attention_all**: All attention projections ($W_q$, $W_v$, $W_k$, $W_o$)
- **mlp_all**: All MLP layers ($W_{gate}$, $W_{up}$, $W_{down}$)
- **all_layers**: All attention and MLP layers combined

**Configuration Selection Algorithm.** Given a set of $k$ target parameter counts $\{d_1, d_2, \ldots, d_k\}$ distributed uniformly in log scale from $d_{min}$ to $d_{max}$ (c.f. Section 2.3), we select configurations as follows:

1. For each target parameter count $d_i$:
   (a) For each module group $g \in \{$attention_q_v, attention_all, mlp_all, all_layers$\}$:
       - Calculate parameters per rank: $\alpha_g = \text{params}(1, g)$
       - Estimate required rank: $r_{est} = \lfloor d_i/\alpha_g \rfloor$
       - Clip rank to valid range: $r = \max(1, \min(r_{est}, d_{model}))$
       - Calculate actual parameters: $d_{actual} = \text{params}(r, g)$
       - Compute error: $\epsilon = |d_{actual} - d_i|$
   (b) Select configuration $(r^*, g^*)$ that minimizes $\epsilon$
2. Store configuration with actual parameter count $d_{actual}$, this naturally handles collisions by keeping only one configuration per unique parameter count.

**Hyperparameter Selection and Convergence.** To ensure the robustness of our training configuration, we conducted preliminary experiments to rigorously determine the optimal training duration and learning rate. We extended training runs up to 15,000 steps to empirically identify the point of convergence, observing that training accuracy and loss consistently plateaued well before our selected limits (8,000 steps for 1B and 6,000 steps for 4B). Additionally, we performed a comprehensive learning rate sweep over a logarithmic scale ranging from $1 \times 10^{-2}$ to $1 \times 10^{-6}$ (evaluating intermediate steps such as $1 \times 10^{-2}, 5 \times 10^{-3}, 1 \times 10^{-3}, \ldots, 1 \times 10^{-6}$). The final learning rates reported in the main text were selected based on the optimal balance of training stability and validation performance observed during this sweep.

## A.2. Intrinsic Dimension Computation for Gemma-3 1B

Figure 3 visualizes the intrinsic dimension computation for the Gemma-3 1B model, analogous to the 4B model visualization in Figure 2 of Section 4. As with the 4B model, we observe that different reasoning strategies exhibit dramatically different parameter efficiency. Executed PoT crosses the threshold ($\tau = 24.3\%$) at only 1.03M parameters, while less effective strategies like Very Short CoT require substantially more capacity (31.45M parameters). The lower threshold for the 1B model (24.3% vs. 63.0% for 4B) reflects the reduced reasoning capability of the smaller model, as discussed in Section 4. Despite this lower performance ceiling, the relative ordering of strategies by intrinsic dimensionality remains consistent with the 4B model, demonstrating that intrinsic dimensionality captures effectiveness of reasoning strategies independent of the model scale.

# B. Size of Training and Test Splits

## B.1. GSM8K

**Evaluation Splits.** We evaluate all models on six test splits spanning both in-distribution (ID) and out-of-distribution (OOD) settings. The in-distribution evaluation uses the GSM8K test set, which contains 1.32K instances (Cobbe et al.,

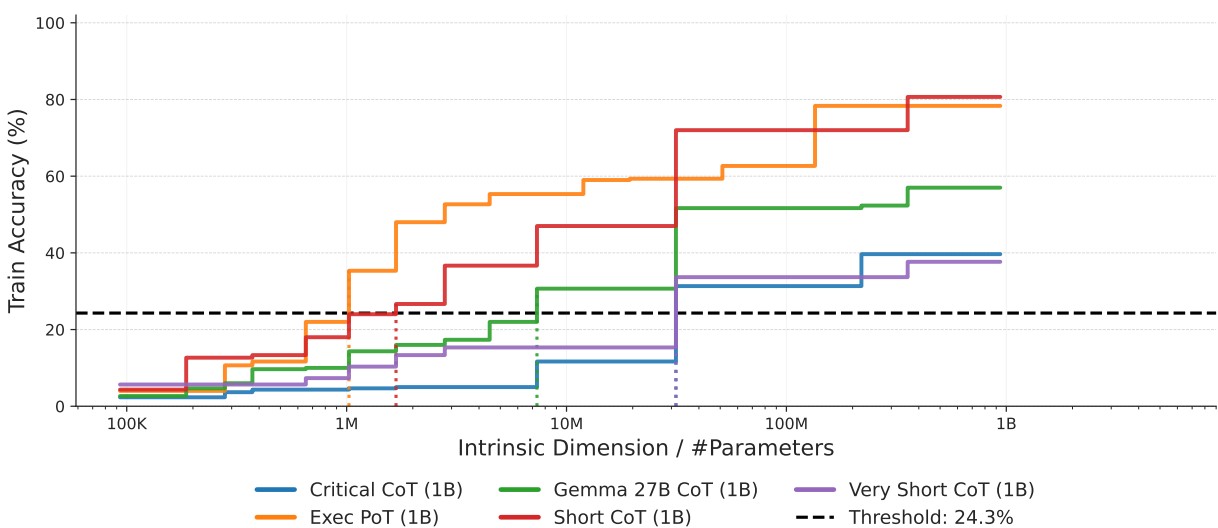

*Figure 3.* Visualization of intrinsic dimension computation for Gemma-3 1B showing select reasoning strategies. We plot the Pareto frontier of monotonic training accuracy versus trainable parameters (log scale). The dashed line indicates the threshold ($\tau = 24.3\%$); intrinsic dimension is the parameter count where each curve first crosses this threshold (vertical dotted lines).

2021).[2] Out-of-distribution evaluations include several GSM-based variants designed to stress different generalization axes: (i) **GSM Symbolic (Main)** (Mirzadeh et al., 2025),[3] consisting of 5K instances generated from distinct symbolic template variations; (ii) **GSM Symbolic P1** (Mirzadeh et al., 2025),[4] a higher-difficulty symbolic split with 5K instances; and (iii) **GSM Symbolic P2** (Mirzadeh et al., 2025),[5] the most challenging symbolic split, containing 2.5K instances. We additionally evaluate on **GSM-IC** (Shi et al., 2023),[6] for which we sample 5K instances from the $m$-step dataset augmented with irrelevant contextual information, and on **GSM-Hard** (Gao et al., 2023),[7] which contains 1.32K instances featuring more challenging arithmetic. Together, these splits enable a systematic assessment of generalization across symbolic structure, difficulty, and robustness to distractors.

**Training Splits across Reasoning Strategies.** As stated in Section 3, we use the GSM8K (Cobbe et al., 2021) training set as the source of questions, which are solved using different reasoning strategies. We prompt a teacher model (Gemini-2.5 or Gemma-3 27B) to generate these solutions using its instruction-following capabilities and filter the outputs for correctness. This yields between 6.2K and 7.5K valid training instances per strategy, with No CoT and No CoT with extra tokens retaining the full 7.5K examples, most reasoning-based CoT variants producing approximately 7.0–7.2K instances, and more constrained formats such as Very Short CoT and Executable PoT resulting in smaller splits (6.7K and 6.2K instances, respectively).Crucially, we find that the variation in training set size *does not* account for the performance differences between strategies. We observe no statistically significant correlation (Spearman Rank) between the number of training samples and downstream accuracy ($\rho \approx -0.11, p > 0.7$ for both 1B and 4B models). Notably, Executed PoT achieves the highest generalization performance despite having the fewest training examples (6.2K), while the No CoT baseline performs poorly despite maximizing the dataset size (7.5K). This confirms that the content of the reasoning data, rather than the number of training examples, are the primary drivers of learnability.

## C. Reasoning Gym

We construct evaluation datasets using Reasoning Gym (Stojanovski et al., 2025), generating four splits: a training set, an in-distribution (ID) test set, a hard OOD test set, and a held-out OOD test set. We describe each below.

[2]Source: https://huggingface.co/datasets/openai/gsm8k/viewer/main/test
[3]Source: https://huggingface.co/datasets/apple/GSM-Symbolic/
[4]Source: https://huggingface.co/datasets/apple/GSM-Symbolic/viewer/p1
[5]Source: https://huggingface.co/datasets/apple/GSM-Symbolic/viewer/p2
[6]Source: https://github.com/google-research-datasets/GSM-IC/blob/main/GSM-IC_mstep.json
[7]Source: https://huggingface.co/datasets/reasoning-machines/gsm-hard

**ID Training and Test Sets.** We generate a large pool of examples using easy task configurations (see Table 6) with a fixed seed. After deduplication based on question text, we apply an 80/20 train/test split, capping training at 3,000 examples per task and the ID test set at 250 examples per task.

**Hard OOD Test Set.** We generate examples using harder configurations of the same tasks (see Table 6), with a different seed to avoid overlap. Examples are decontaminated by removing any question appearing in the training set, and each task is capped at 250 examples.

**Held-Out OOD Test Set.** We evaluate on task types not seen during training. For *cognitive* reasoning, we use `needle_haystack` and `modulo_grid` (250 examples each); for *algorithmic* reasoning, we use `letter_counting` and `count_primes` (250 examples each). All held-out sets are decontaminated against the training set. The **OOD** score in Table 4 is the geometric mean across the hard OOD and held-out OOD sets; **Overall** is the geometric mean across all three test sets (ID, hard OOD, held-out OOD).

*Table 6.* Algorithmic and cognitive reasoning task ask configurations for easy (ID), hard OOD, and held-out OOD splits of Reasoning Gym tasks evaluated in Table 4.

| Task | Group | Easy Config | Hard Config |
|---|---|---|---|
| color_cube_rotation | Cognitive | 1–5 rotations | 8–15 rotations |
| rectangle_count | Cognitive | ≤10 rectangles | ≤15 rectangles |
| number_sequence | Cognitive | 4–8 terms, $|v| \leq 100$, complexity $\leq 3$ | 5–10 terms, $|v| \leq 500$ |
| leg_counting | Algorithmic | 2–10 animals, 1–15 instances | 12–20 animals, 64–256 instances |
| base_conversion | Algorithmic | base 2–16, value 0–1000 | base 9–18, value 2500–10000 |
| rotate_matrix | Algorithmic | $n \in [2, 10]$, 0–10 rotations | $n \in [12, 25]$, 5–15 rotations |
| *Held-Out OOD Tasks (unseen task types)* | | | |
| needle_haystack | Cognitive | Default configuration | |
| modulo_grid | Cognitive | 20×20 grid, max divisor 20, max target 20 | |
| letter_counting | Algorithmic | 5–15 words | |
| count_primes | Algorithmic | $n \in [1, 1000]$ | |

## D. Comparison with LongPPL

Token perplexity is length-normalized and treats all tokens equally regardless of position. LongPPL (Fang et al., 2025) addresses this by down-weighting tokens in the short-context prefix and assigning higher weight to tokens beyond position $K$, using a long-short context contrastive method. It is primarily designed for long-context evaluation settings. We evaluate two configurations following the original $\alpha$ and $\beta$ values from Fang et al. (2025): $K$=64 and $K$=256. As shown in Table 7, both LongPPL variants achieve lower Spearman correlations (0.60 and 0.62) than regular token perplexity (0.82) on Gemma-3 4B, and fall well short of intrinsic dimensionality (0.93). Since token perplexity is already length-normalized, it is less sensitive to sequence length than KL divergence; the CoT strategies we evaluate are well within the short-context window for both $K$ values, which may explain why the additional long-context correction does not provide further gains in our setting.

## E. Examples of Different Reasoning Strategies

Below are examples of the reasoning strategies evaluated in this work. All examples use the same base question to illustrate the differences in generation structure.

*Table 7.* Spearman rank correlations with generalization performance (Gemma-3 4B) for PPL-based metrics and intrinsic dimensionality. LongPPL variants use the original $\alpha, \beta$ values from Fang et al. (2025).

| Metric | Spearman Correl. ($\uparrow$) |
|---|---|
| LongPPL ($K{=}64$) | 0.60 |
| LongPPL ($K{=}256$) | 0.62 |
| Token Perplexity | 0.82 |
| Intrinsic Dimensionality **(ours)** | **0.93** |

---

### No CoT (Sprague et al., 2025)

*Outputs a direct answer without intermediate reasoning.*

**Question:** Natalia sold clips to 48 of her friends in April, and then she sold half as many clips in May. How many clips did Natalia sell altogether in April and May?

**Model Output:** Answer: 72.

---

### No CoT with Extra Tokens

*Appends filler text to isolate the effect of inference-time computation from reasoning quality.*

**Question:** Natalia sold clips to 48 of her friends in April, and then she sold half as many clips in May. How many clips did Natalia sell altogether in April and May?

**Model Output:** Thinking about the answer to this math word problem and formatting it ...

Answer: 72.

---

### Very Short CoT (Nye et al., 2022)

*Prompted for concise, equation-style reasoning.*

**Question:** Natalia sold clips to 48 of her friends in April, and then she sold half as many clips in May. How many clips did Natalia sell altogether in April and May?

**Model Output:** April sales plus May sales (48 + 48/2) equals total sales. Answer: 72

---

### Short CoT

*Restricted to brief (1–2 sentence) explanations.*

**Question:** Natalia sold clips to 48 of her friends in April, and then she sold half as many clips in May. How many clips did Natalia sell altogether in April and May?

**Model Output:** Natalia sold half of 48 clips in May, which is $48/2 = 24$ clips, and then added that to the 48 clips sold in April for a total of $48 + 24 = 72$ clips. Answer: 72

## Short CoT with $n = 4$ Distractors ([Li et al., 2025](#))

*Unrelated steps sampled from other problems are inserted before reaching the correct answer to test robustness.*

**Question:** Natalia sold clips to 48 of her friends in April, and then she sold half as many clips in May. How many clips did Natalia sell altogether in April and May?

**Model Output:** Let Chase's speed be $c$, Cameron's speed be $2c$, and Danielle's speed be $3(2c) = 6c$. Since time = distance/speed, and the distance is the same for all three, we can set up a proportion: 30 minutes / $(6c) = t/c$, where $t$ is the time it takes Chase to travel the distance; solving for $t$, we get $t = 30 \times c/(6c) = 5 \times 30 = 180$ minutes.

But wait, this is wrong. Let me try again.

First, we find Joshua made $20 + 6 = 26$ macarons, and Miles made $26 \times 2 = 52$ macarons; then, Renz made $(3/4) \times 52 - 1 = 39 - 1 = 38$ macarons, so the total number of macarons is $20 + 26 + 52 + 38 = 136$, and $136/2 = 68$ kids would receive 2 macarons each.

But wait, this is wrong. Let me try again.

First, subtract the unripe apples from the total to find the number of ripe apples: $34 - 6 = 28$ ripe apples. Then, divide the number of ripe apples by the number of apples needed per pie to find the number of pies they can make: $28/4 = 7$ pies.

But wait, this is wrong. Let me try again.

First, calculate the total points needed to tie the record: 287 points/player/round $\times$ 4 players $\times$ 10 rounds = 11,480 points. Then, determine the points needed in the final round: $11,480 - 10,440 = 1,040$ points, and divide by the number of players to find the required average: $1,040/4 = 260$ points/player. Finally, find the difference between the record average and the needed average: $287 - 260 = 27$ points.

But wait, this is wrong. Let me try again.

Natalia sold half of 48 clips in May, which is $48/2 = 24$ clips, and then added that to the 48 clips sold in April for a total of $48 + 24 = 72$ clips. Answer: 72

## Gemma 27B CoT

*Allows unconstrained reasoning (generated by Gemma-3 27B).*

**Question:** Natalia sold clips to 48 of her friends in April, and then she sold half as many clips in May. How many clips did Natalia sell altogether in April and May?

**Model Output:** Let the number of clips Natalia sold in April be $A$. Let the number of clips Natalia sold in May be $M$. We are given that Natalia sold clips to 48 of her friends in April, so $A = 48$. We are also given that she sold half as many clips in May as she did in April, so $M = \frac{1}{2}A$. Since $A = 48$, we have $M = \frac{1}{2}(48) = 24$. The total number of clips Natalia sold in April and May is $A + M = 48 + 24$. $48 + 24 = 72$. Therefore, Natalia sold 72 clips altogether in April and May. Answer: 72

## Plan and Solve (Wang et al., 2023b)

*Follows a decomposition framework to plan before executing.*

**Question:** Natalia sold clips to 48 of her friends in April, and then she sold half as many clips in May. How many clips did Natalia sell altogether in April and May?

**Model Output:** Here's a step-by-step plan to solve the problem:

1. **Find the number of clips sold in May:** The problem states Natalia sold half as many clips in May as she did in April. We need to calculate half of the number of clips sold in April. 2. **Find the total number of clips sold:** Add the number of clips sold in April to the number of clips sold in May to find the total number of clips sold over the two months.

Now, let's execute the plan:

**Step 1: Find the number of clips sold in May**

- Natalia sold 48 clips in April.

- She sold half as many in May, so we need to calculate 48 / 2.

- $48/2 = 24$ clips

**Step 2: Find the total number of clips sold**

- Clips sold in April: 48

- Clips sold in May: 24

- Total clips sold: $48 + 24 = 72$

Answer: 72

## Executed PoT (Gao et al., 2023; Chen et al., 2023)

*Uses actual program execution to derive the answer.*

**Question:** Natalia sold clips to 48 of her friends in April, and then she sold half as many clips in May. How many clips did Natalia sell altogether in April and May?

**Model Output:**

```
def solution():
    # Natalia sold clips to 48 friends in April.
    april_sales = 48
    # She sold half as many clips in May.
    may_sales = april_sales / 2
    # Calculate the total sales in April and May.
    total_sales = april_sales + may_sales
    return int(total_sales)
```

## Simulated PoT ([Sprague et al., 2025](#))

*Relies on internal code simulation (reasoning about code) rather than external execution.*

**Question:** Natalia sold clips to 48 of her friends in April, and then she sold half as many clips in May. How many clips did Natalia sell altogether in April and May?

**Model Output:**

```
# Natalia sold clips to 48 friends in April.
april_sales = 48
# She sold half as many clips in May.
may_sales = april_sales / 2
# Calculate the total sales in April and May.
total_sales = april_sales + may_sales
# Print the total sales.
print(total_sales)
```

**Simulation of Execution:**

1. `april_sales` is assigned the value 48.
2. `may_sales` is calculated as `april_sales / 2`, which is $48/2 = 24$.
3. `total_sales` is calculated as `april_sales + may_sales`, which is $48 + 24 = 72$.
4. The value of `total_sales` (72) is printed.

Answer: 72

## High Review Ratio CoT ([Feng et al., 2025](#))

*Features higher occurrences of revision tokens for longer and verification-based reasoning.*

**Question:** Natalia sold clips to 48 of her friends in April, and then she sold half as many clips in May. How many clips did Natalia sell altogether in April and May?

**Model Output:** Okay, let's break down this problem step-by-step. Natalia sold clips to 48 friends in April. In May, she sold half as many. We need to find the total number of clips sold in both months.

First, let's find out how many clips she sold in May. Half of 48 is $48/2$. Let me double check that division. 48 divided by 2 is indeed 24. So, she sold 24 clips in May.

Now, we need to find the total number of clips sold in April and May. That means we need to add the number of clips sold in April (48) to the number of clips sold in May (24). So, $48 + 24$. Let's verify this addition. $48 + 20 = 68$, and $68 + 4 = 72$.

Therefore, Natalia sold a total of 72 clips in April and May. Wait, the problem states she sold *clips to* her friends. Does that mean each friend bought one clip? It seems reasonable to assume that each friend purchased one clip, so the number of friends is equal to the number of clips sold. Let me verify this assumption. The problem doesn't explicitly state this, but it's the most logical interpretation.

Let's go back and review the calculations. April: 48 clips. May: $48/2 = 24$ clips. Total: $48 + 24 = 72$ clips. I should check the addition one more time to be absolutely sure. $48 + 24 = 72$. Yes, that's correct.

Answer: 72

### Critical CoT (Zhou et al., 2024)

*Uses a reasoning structure associated with critical-thinking strategies (deconstruction, questioning assumptions).*

**Question:** Natalia sold clips to 48 of her friends in April, and then she sold half as many clips in May. How many clips did Natalia sell altogether in April and May?

**Model Output:** Okay, let's break down this problem using both critical and reflective thinking.

#### 1. Understanding the Problem (Critical Thinking - Deconstruction)

The problem states Natalia sold clips to a specific number of friends in April (48). It then tells us she sold a *relative* amount in May – half as many as in April. The question asks for the *total* number of clips sold across both months. The key here is understanding "half as many" and recognizing that we need two steps: calculate May's sales, then add April and May's sales together.

#### 2. Identifying Assumptions (Critical Thinking - Questioning)

- **Assumption:** The problem implies Natalia sold *one* clip to each friend. This is a crucial assumption. If she sold multiple clips to each friend, the problem would be unsolvable without more information. We'll proceed *assuming* one clip per friend.
- **Assumption:** The problem is dealing with whole numbers of clips. It's unlikely Natalia sold half a clip.

#### 3. Solving the Problem (Critical Thinking - Logical Reasoning)

- **Step 1: Calculate May's sales.**
  May's sales = April's sales / 2
  May's sales = $48/2 = 24$
- **Step 2: Calculate total sales.**
  Total sales = April's sales + May's sales
  Total sales = $48 + 24 = 72$

#### 4. Reflective Thinking - Checking for Reasonableness & Alternative Approaches

- **Reasonableness Check:** Does the answer make sense? Natalia sold almost 50 clips in April, and roughly half that in May. A total of 72 seems plausible.
- **Alternative Approach:** We could have thought of May's sales as a fraction of the total. If May is 1/2 of April, then May is 1 part and April is 2 parts... total sales is $3 \times 24 = 72$. This confirms our initial approach.

#### 5. Final Answer

Answer: 72

---

**Gemini CoT (Gemini 2.5 Flash; Comanici et al., 2025)**

*Produced by a stronger teacher model known for longer solutions.*

**Question:** Natalia sold clips to 48 of her friends in April, and then she sold half as many clips in May. How many clips did Natalia sell altogether in April and May?

**Model Output:** Here's my thinking process to solve this problem:

1. **Understand the Goal:** The question asks for the *total* number of clips sold in April *and* May.

2. **Break Down the Information:**

   - Clips sold in April: 48
   - Clips sold in May: Half as many as April.

3. **Calculate Clips Sold in May:**

   - "Half as many as April" means April's number divided by 2.
   - May clips $= 48/2$
   - May clips $= 24$

4. **Calculate Total Clips Sold:**

   - Total = Clips in April + Clips in May
   - Total $= 48 + 24$

5. **Perform the Addition:**

   - $48 + 24 = 72$

6. **Formulate the Answer:** Present the steps clearly and state the final answer in the requested format.

Here's how to solve the problem:

1. **Clips sold in April:** Natalia sold 48 clips in April.

2. **Clips sold in May:** She sold half as many clips in May as in April. To find half, we divide by 2: $48$ clips$/2 = 24$ clips

3. **Total clips sold:** To find the total, we add the clips sold in April and May: $48$ clips (April) $+ 24$ clips (May) $= 72$ clips

Answer: 72

---

## F. Detailed Results across all Test Splits

*Table 8.* Detailed Performance of Gemma-3 4B across test splits. **Symb**: GSM-Symbolic; **P1/P2**: Symbolic Perturbations; **IC**: GSM-IC; **Hard**: GSM-Hard. **OOD**: Geometric Mean of the 5 stress tests. **Overall**: Geometric Mean of all 6 splits.

| | ID | Out-of-Distribution (Stress Tests) | | | | | Aggregates | |
|---|---|---|---|---|---|---|---|---|
| **CoT Strategy** | **GSM8K** | **Symb** | **P1** | **P2** | **IC** | **Hard** | **OOD** | **Overall** |
| *Baseline Strategies* | | | | | | | | |
| No CoT | 14.94 | 7.76 | 3.70 | 2.25 | 34.62 | 4.09 | 6.20 | 7.18 |
| No CoT with extra tokens | 16.45 | 7.64 | 4.16 | 2.96 | 48.64 | 4.09 | 7.15 | 8.22 |
| *Short CoT Variants* | | | | | | | | |
| Very Short CoT | 44.58 | 33.00 | 16.70 | 6.12 | 53.64 | 14.48 | 19.22 | 22.11 |
| Short CoT | 58.98 | 49.36 | 28.54 | 10.80 | 66.70 | 20.17 | 28.99 | 32.63 |
| Short CoT with 2 Distractors | 50.11 | 39.32 | 19.90 | 7.24 | 66.72 | 17.36 | 23.09 | 26.27 |
| Short CoT with 4 Distractors | 41.32 | 31.74 | 15.98 | 5.48 | 56.28 | 14.78 | 18.74 | 21.38 |
| Short CoT with 8 Distractors | 22.97 | 18.90 | 8.22 | 2.64 | 33.02 | 8.19 | 10.21 | 11.69 |
| *Default CoTs Sampled from Teacher Model* | | | | | | | | |
| Gemma 27B CoT | 67.48 | 56.82 | 42.20 | 16.36 | 78.56 | 26.54 | 38.24 | 42.04 |
| Gemini CoT | 66.72 | 56.10 | 39.60 | 15.32 | 66.66 | 24.72 | 35.46 | 39.40 |
| *Specific Reasoning Strategies from Prior Work* | | | | | | | | |
| Executed PoT | 62.77 | 63.64 | 48.24 | 18.80 | 63.30 | 42.15 | **43.40** | **46.15** |
| Simulated PoT | 64.75 | 53.80 | 39.74 | 15.08 | 63.40 | 26.16 | 35.13 | 38.90 |
| Plan Solve | 64.75 | 54.54 | 36.06 | 13.52 | 71.64 | 25.02 | 34.33 | 38.16 |
| Critical CoT | 63.84 | 55.72 | 37.98 | 13.84 | 68.34 | 21.83 | 33.74 | 37.52 |
| High Review Ratio CoT | **67.63** | 61.44 | 42.84 | 19.16 | 78.40 | 26.23 | 40.10 | 43.75 |

*Table 9.* Detailed Performance of Gemma-3 1B across test splits. **Symb**: GSM-Symbolic; **P1/P2**: Symbolic Perturbations; **IC**: GSM-IC; **Hard**: GSM-Hard. **OOD**: Geometric Mean of the 5 stress tests. **Overall**: Geometric Mean of all 6 splits.

| | ID | Out-of-Distribution (Stress Tests) | | | | | Aggregates | |
|---|---|---|---|---|---|---|---|---|
| **CoT Strategy** | **GSM8K** | **Symb** | **P1** | **P2** | **IC** | **Hard** | **OOD** | **Overall** |
| *Baseline Strategies* | | | | | | | | |
| No CoT | 3.56 | 0.90 | 0.94 | 0.80 | 17.64 | 0.91 | 2.00 | 1.84 |
| No CoT with extra tokens | 5.31 | 1.50 | 1.18 | 1.92 | 7.08 | 1.44 | 2.00 | 2.38 |
| *Short CoT Variants* | | | | | | | | |
| Very Short CoT | 8.95 | 4.32 | 2.14 | 1.20 | 39.82 | 1.82 | 4.00 | 4.39 |
| Short CoT | 18.04 | 9.96 | 3.98 | 1.92 | 72.08 | 4.32 | 7.00 | 8.68 |
| Short CoT with 2 Distractors | 10.46 | 6.76 | 1.98 | 1.40 | 79.04 | 1.67 | 5.00 | 5.44 |
| Short CoT with 4 Distractors | 4.78 | 2.70 | 1.16 | 0.48 | 65.04 | 1.29 | 3.00 | 2.91 |
| Short CoT with 8 Distractors | 2.43 | 1.00 | 0.66 | 0.88 | 45.60 | 0.23 | 1.00 | 1.57 |
| *Default CoTs Sampled from Teacher Model* | | | | | | | | |
| Gemma 27B CoT | 20.40 | 11.54 | 3.50 | 2.04 | 44.74 | 4.25 | 7.00 | 8.27 |
| Gemini CoT | 20.55 | 12.66 | 5.38 | 2.12 | 64.90 | 4.40 | 8.00 | 9.73 |
| *Specific Reasoning Strategies from Prior Work* | | | | | | | | |
| Executed PoT | 20.24 | 18.54 | 8.80 | 1.96 | 38.28 | 10.69 | **11.00** | **11.76** |
| Simulated PoT | 20.85 | 9.74 | 5.64 | 1.84 | 59.58 | 4.17 | 8.00 | 8.98 |
| Plan Solve | 21.53 | 13.86 | 6.78 | 2.88 | 74.86 | 4.62 | 10.00 | 11.24 |
| Critical CoT | 17.51 | 11.98 | 4.62 | 2.28 | 57.96 | 4.47 | 8.00 | 9.11 |
| High Review Ratio CoT | **22.60** | 14.30 | 5.84 | 2.20 | 66.28 | 5.08 | 9.00 | 10.57 |

