# OpenReview forum: "Effective Reasoning Chains Reduce Intrinsic Dimensionality"
_ICML.cc/2026/Conference — ICML 2026 spotlight_

### Official Review · Reviewer_PqZG · 2026-03-10

**Soundness:** 2
**Presentation:** 2
**Significance:** 2
**Originality:** 2
**Overall Recommendation:** 4
**Confidence:** 3

**Summary:**

Current explanations for why Chain-of-thought improves model performance mostly attribute it to increased test-time computation or structural guidance. The authors propose intrinsic dimensionality to measure the parameter efficiency of different reasoning strategies, and define it as the minimum number of trainable parameters required, under a fixed training setup, for the model to reach a preset accuracy threshold during training. Based on the GSM8K dataset, the authors construct data variants for different reasoning strategies, and fine-tune Gemma 3 1B and 4B models with LoRA. The authors compare the model accuracy under different scales of trainable parameters, and accordingly compute the minimum number of trainable parameters required to reach the threshold $\tau$. The experimental results show that more effective reasoning strategies often correspond to smaller intrinsic dimensionality (i.e., higher parameter efficiency).

**Compliance With Llm Reviewing Policy:**

Affirmed.

**Final Justification:**

The authors have adequately addressed my concerns in their rebuttal, and the additional experiments they provided sufficiently demonstrate the generality.

**Key Questions For Authors:**

1. In Appendix A.1, the authors use “selecting the configuration that minimizes the absolute error between the target parameter count and the actual parameter count” to address the issue that “different combinations of LoRA target modules and rank may yield similar parameter counts but different performance.” What is the reason for this, and why can it solve this problem?
2. What factors mainly lead to the differences in intrinsic dimensionality corresponding to different reasoning strategies?
3. If one tries training on Chain-of-thought datasets generated by different models, will the ordering relationship of intrinsic dimensionality still be consistent?

**Limitations:**

The current experimental coverage is relatively narrow (limited models and datasets), and the mechanistic explanation behind the phenomena is still insufficient.

**Strengths And Weaknesses:**

Strengths
1. This work attempts to reconcile differing conclusions in the literature regarding the CoT mechanism, and some of its experimental observations are broadly consistent with prior studies, making it an empirical validation of existing intuitions.
2. The method setup is relatively straightforward, which facilitates reproduction and verification.

Weaknesses
1. The experimental coverage is limited in terms of model scale and dataset diversity, and further empirical validation is needed.
2. Several key definitions and figure explanations are not sufficiently clear. For example, the definition of the threshold \tau is unclear, and the description of the relationship between “Middle (green)” and $\tau$ in Figure 1 is somewhat ambiguous.
3. Overall, the work is more of an empirical validation of existing intuitions, and its conceptual novelty appears relatively limited.

---

> ### Author Rebuttal · Authors · 2026-03-31
>
> We thank the reviewer for their careful reading of our work. We note that several of the points raised, in particular around experimental coverage, are directly addressed by additional experiments conducted during the rebuttal period. We address each point in turn.
>
> ---
>
> >\[W1 / Q3\] Limited model scale and dataset diversity; different teacher models.
>
> Your point is well-taken and we have attempted to address it through additional experiments. We conducted experiments on Reasoning Gym (Stojanovski et al., 2025\) across cognitive and algorithmic reasoning categories, demonstrating that the strong inverse correlation between intrinsic dimensionality and generalization holds beyond GSM8K (see response to **Reviewer eRSy** for full details). Regarding teacher model consistency, we point the reviewer directly to Table 1 (L233-235): Gemma 27B CoT and Gemini CoT, generated by meaningfully different teacher models, and find the inverse relationship between generalization performance and intrinsic dimensionality still holds. The same is true for Gemma 1B in Table 2\. We will expand both analyses in the final version.
>
> ---
>
> >\[W2\] Key definitions and figure explanations are not sufficiently clear.
>
> Regarding **τ**: we define it formally in Section 2.3 as the maximum training accuracy achieved by any strategy at full capacity after one epoch, used as a common threshold across all strategies. We will add a more explicit self-contained definition earlier in the section. Regarding **Figure 1**: the "Middle (green)" panel illustrates the intrinsic dimension computation for a single reasoning strategy, showing how we sweep over LoRA configurations of varying parameter counts d and identify the minimum d crossing τ. This is analogous to Figure 2, which shows the same Pareto frontier across multiple strategies simultaneously. We will revise the caption to make this connection explicit.
>
> ---
>
> >\[W3\] Limited conceptual novelty; empirical validation of existing intuitions.
>
> We respectfully push back on this characterization. As discussed in lines 407-432, prior work has proposed qualitative intuitions about CoT effectiveness but these either lack quantifiability or yield conflicting empirical evidence, for example the relationship between trajectory length and performance shows inconsistent results (Wu et al., 2025; Marjanovic et al., 2025), as noted in lines 25-54. Our contribution is to establish intrinsic dimensionality as a principled, quantitative, and consistently predictive metric for comparing reasoning strategies, grounded in the Minimum Description Length principle, outperforming existing alternatives with a Spearman correlation of 0.93 on 4B and 0.75 on 1B. This is not merely a validation of prior intuitions but a new measurement framework that reconciles conflicting prior findings.
>
> ---
>
> >\[Q1\] On LoRA configurations and the greedy selection procedure.
>
> As described in Appendix A.1 (lines 605-632), different combinations of LoRA rank and target modules drawn from four groups (`attention_q_v`, `attention_all`, `mlp_all`, `all_layers`) can yield similar parameter counts. By greedily selecting the configuration minimizing the absolute error between target and actual parameter count for each grid point, we ensure the sweep remains uniform in log space and each grid point is represented by the most precisely matched configuration. Crucially, since this algorithm is applied identically across all reasoning strategies, any residual structural bias impacts all strategies equally, preserving the relative ordering that drives our results.
>
> ---
>
> >\[Q2\] What factors mainly lead to differences in intrinsic dimensionality across reasoning strategies?
>
> The core driver is how effectively a reasoning strategy bridges the logical gap between input and answer. Strategies such as Executed PoT impose a highly structured, executable mapping that is readily compressible with few parameter updates. In contrast, in case of No CoT baseline, without the ability to unroll complex computations, or in case of strategies with irrelevant noise (Short CoT with distractors), the model might resort to memorizing individual solutions requiring more capacity and leading to reduced generalization.
>
> ---
>
> We hope these responses, together with our new experimental results, directly address the reviewer's concerns around experimental coverage, and clarity, and allow them to revisit their overall assessment. We are happy to address any remaining questions.

---

> > ### Author Rebuttal · Reviewer_PqZG · 2026-04-01
> >
> > Thank you to the authors for their response and for adding the relevant experiments, which have addressed my concerns. I will raise my score accordingly.

---

> > > ### Author Response · Authors · 2026-04-07
> > >
> > > Thank you for your helpful suggestions and for increasing the score. We are glad you appreciated our additional experiments and will include them in the final version of the paper.

---

### Official Review · Reviewer_iCuf · 2026-03-11

**Soundness:** 3
**Presentation:** 3
**Significance:** 3
**Originality:** 4
**Overall Recommendation:** 5
**Confidence:** 3

**Summary:**

In the case of solving simple math problems, the paper finds that LLMs, fine-tuned to reach a certain accuracy in training, that need less parameters to do this, have a better in-test accuracy than LLMs that need more parameters. The paper does this comparison between 14 models that are trained to solve the math problem with a given CoT strategy, from no-CoT to PoT, so the result is restricted to this setting. The interpretation is that if a model is forced to use less parameters to solve math problems, it is incentivised to find really math solutions, that are usually compact; while if the model is allowed to use more parameters, it degenerates more easily to find other reasoning paths that do not generalise well. So authors interpret their results as a sign that reasoning in lower dimensions is better. To come to this conclusions, an interesting experiment is set up. For each of the different reasoning strategies, a LORA is fine tuned with d parameters, with d varying on a grid. Then the smallest d is determined for which a strategy reaches a given level of accuracy in training. This is called intrinsic dimension of the reasoning strategy. Finally, in domain and out of domain testing the accuracy of this strategy is computed. A series of investigations are presented to support their findings.

**Compliance With Llm Reviewing Policy:**

Affirmed.

**Final Justification:**

I think the authors clarified all my doubts, and replied to my criticism in a convincing way. I also take into account the response to eRSy, which shows that their idea is useful beyond maths benchmarks. it is a clever construction.

**Key Questions For Authors:**

1, Can you justify your conclusion that "findings provide a principled, quantitative explanation for why different reasoning strategies
improve generalization"?

2. What is the correlation with token perplexity?

3. Is this an effect only on easier problems? What happens if you take cases where your tau is much lower?

4. Have you repeated the runs, so that in fact you have multiple training for each d of PoT (and the others)? In this case you would have a confidence interval for the intrinsic dimension. This uncertainty would mirror in a confidence interval for the correlation in Table 3 and in plots.

5. You need a dimension, and use LORA. You could use other forms of complexity, including just the number of actual parameters of the LLMs. Would that work? If so what would you expect?

6. I wonder what would happen if you would use instances where the result is wrong in training. This would fine-tune the model to reason wrongly. What happens then to the relation between intrinsic dimension and test-accuracy? Here you should not see the high correlation. I am looking for experiments that support your interpretation further.

7. Are you using important and contemporary LLMs?

**Limitations:**

This is for a very special type of problems and definition of dimensionality. But still imtersting case!

**Strengths And Weaknesses:**

I would like to challenge the interpretation that "findings provide a principled, quantitative explanation for why different reasoning strategies
improve generalization", the reason being: their intrinsic dimension. First there is no evidence of such a generality. Second, I would expect a discussion about confounders: is the intrinsic dimension a feature that is correlated to something else and it is this something else that really causally matters? What could this else be, or why there is no such? What is the correlation with token perplexity? I would like to see this more discussed to get convinced about full soundness.

The relation to information theory is far fetched, and not particularly useful: you have a strong defintion of intrinsic dimension, that stands on its own legs.

A good paper, with a novel pipeline and interesting results.

---

> ### Author Rebuttal · Authors · 2026-03-31
>
> We thank the reviewer for their overall positive assessment: "a good paper, with a novel pipeline and interesting results." Below, we provide specific responses to each of your points.
>
> ---
>
> >\[W1 / Q1 / Q2\] Potential confounders; relationship with token perplexity needs more discussion.
>
> On **confounders and token perplexity** (defined in L177-187, discussed in L265-303): we compute a Spearman rank correlation of 0.91 (4B) and 0.50 (1B) between ID and token perplexity across strategies, suggesting they are related but not equivalent, particularly for smaller models. This is consistent with our interpretation that reasoning chains with high likelihood under the base model tend to also be more compressible (Yue et al., 2025). The weaker alignment on the 1B model suggests intrinsic dimensionality captures additional signal, perhaps reflecting global coherence across the full set of training trajectories rather than per-instance aggregation.
>
> On **generality**: as motivated in L25-54 and L407-432, existing approaches to explaining CoT effectiveness are either not quantifiable or yield conflicting empirical evidence. Intrinsic dimensionality consistently achieves Spearman correlations of 0.93 (4B) and 0.75 (1B), outperforming all alternatives across 14 diverse reasoning strategies. Our Reasoning Gym experiments (see response to **Reviewer eRSy**) further demonstrate this holds on qualitatively different domains.
>
> In the revision, we will moderate the term "explanation" to better reflect that this is a principled quantitative predictor, and will explicitly discuss limitations with respect to establishing the causal as opposed to simply predictive nature of the relationship between intrinsic dimensionality and generalization.
>
> ---
>
> >[Q3] What happens if tau is much lower?
>
> Section 4.3 and Table 3 directly address this: strong correlations persist across thresholds from 70% to 90% of maximum epoch-1 training accuracy. Prior work (Aghajanyan et al., 2021) exclusively used 90% of validation accuracy; we explored a wider range to understand sensitivity to this choice. τ reflects the target model's own capabilities, with thresholds for 1B (τ=24.3%) and 4B (τ=63.0%) reflecting their respective performance ceilings (Figures 2, 3, Appendix A.2). Values significantly below 70% may not meaningfully differentiate strategies given the granularity of our LoRA sweep.
>
> ---
>
> >[Q4] Have you repeated runs to obtain confidence intervals?
>
> Due to the computational cost and rebuttal time constraints, we cannot provide confidence intervals for all settings. We ran 3 independent runs for a representative subset on the 4B model (No CoT, Short CoT, Gemma 27B CoT, Gemini CoT, High Review Ratio CoT): No CoT shows the highest variance at ±3.1 points on overall accuracy, while all other CoT variants fall between ±0.6 and ±1.9 points. Crucially, rank ordering of strategies did not change across runs, which is the key quantity driving our Spearman correlations. We already apply smoothing to training accuracy curves and will include formal uncertainty estimates in the final version.
>
>
> ---
>
> >\[Q5\] Could you use other forms of complexity, including just the number of actual parameters of the LLM?
>
> Naive measures of base LLM complexity such as parameter count are actually inversely correlated with generalization in our setting: larger models exhibit lower intrinsic dimensionality for effective reasoning strategies (Tables 1, 2, Sec 4.4), consistent with Aghajanyan et al. (2021). Intrinsic dimensionality is more useful precisely because it captures the *conditional complexity of the task given the pre-trained model*, varying meaningfully across reasoning strategies even when total capacity is held fixed. Token perplexity and KL divergence serve as alternative measures of this conditional complexity, though they correlate less reliably with generalization. We leave further exploration to future work.
>
> ---
>
> >\[Q6\] What would happen if you fine-tuned on instances where the training answer is wrong?
>
> We thank the reviewer, their intuition is correct. We ran this experiment on Gemma-3 4B using Short CoT and Gemma 27B CoT, constructing training sets with 50% correct and 50% incorrect responses. Mixing in incorrect reasoning chains leads to both ***higher intrinsic dimensionality and lower generalization performance***, consistent with our interpretation that incorrect reasoning does not provide a consistent, compressible input-output mapping, requiring more parameters to fit. The results are summarized below:
>
> | CoT Strategy  | Overall Acc. | Intrinsic Dimension (M) |
> | :---- | :---- | :---- |
> | Short CoT (100% correct) | 32.63 | 3.92 |
> | Short CoT (50% correct) | 20.61 | 2729.64 |
> | Gemma 27B CoT (100% correct) | 42.04 | 2.05 |
> | Gemma 27B CoT (50% correct) | 29.24 | 257.18 |
>
> We will include this ablation in the final version of the paper.
>
> ---
>
> We hope our response directly addresses the reviewer's points and allows them to revisit their score.

---

> > ### Author Rebuttal · Reviewer_iCuf · 2026-04-02
> >
> > Clear answers to all my points. Only residual thought on W1/Q1: I still think that you should discuss, and possibly have evidence to exclude as you write here, the possibility that ID is a confounder. I will increase my score.

---

> > > ### Author Response · Authors · 2026-04-02
> > >
> > > Thank you for your positive acknowledgement and for agreeing to raise your score. Your point on W1/Q1 is well-taken and we want to assure you that we will include a dedicated discussion around the possibility of intrinsic dimensionality being a confounder in the final version, including evidence where available to either support or exclude this.

---

### Official Review · Reviewer_pBeX · 2026-03-12

**Soundness:** 3
**Presentation:** 3
**Significance:** 2
**Originality:** 3
**Overall Recommendation:** 4
**Confidence:** 3

**Summary:**

In this paper, the authors identify intrinsic dimensionality as a quantitative measure for characterizing the effectiveness of reasoning chains, i.e., CoTs.  Specifically, they demonstrate that effective reasoning strategies consistently reduce the intrinsic dimensionality of the task across different reasoning strategies. Their experiments are conducted on GSM8K for both Gemma-3 1B and 4B
models on in-distribution and out-of-distribution evaluations.

**Compliance With Llm Reviewing Policy:**

Affirmed.

**Final Justification:**

The authors have addressed my main concern, about the training-induced generalisability, and added the comparison with other CoT-quality measurements.

A comprehensive literature review, covering both training-free and training-required methods, is strongly encouraged to be included in the next version.

**Key Questions For Authors:**

1.  From the section 2, Intrinsic dimensionality (ID) 's calculation is based on the minimum number of trainable parameters. How to derive it effectively?  Do we need to train the model first and then derive the ID. If so, any alternative way of deriving it before training.  In other words, if the training actually change the model's perspective to the CoTs, more like post-hoc measurement.

2.  How about PPL variant, such as LongPPL? as PPL is the second-performing method, so I wonder the performance of its advanced version and it is training free.

3. Any other perspectives, such as faithfulness can be good metric of CoT quality. A more comprehensive comparisons (related work) ?

**Limitations:**

The method requires fine-tuning, has been mentioned in the conclusion and discussion.

**Strengths And Weaknesses:**

Strengths:
- as far as I understand, it is the first paper to analyse the CoT's effectiveness via intrinsic dimensions and it provides empirical evidence to support the claim.
- The paper is well-written and easy-to-understand

Weakness:
- The paper has shown the correlation between reasoning effectiveness and CoT's related characteristics, i.e., TokenPPL, length, KL divergence, but these aspects are kinda too simple. For example, PPL is actually easily to be perturbed by the length, so, more baselines about CoT's features are encouraged, as well as the literature review in CoT related analysis should cover. such as [1].
- Evaluation models are datasets are relatively limited, only one dataset, GSM8k, and the Gemma models.
- this method requires fine-tuning on new dataset, so not sure whether training on different dataset will change the ID measurement for the same CoTs. In that way, it is not motivated what the ID works for as it isn't an intrinsic characteristics (internal) of the model, but triggered by (variant) datasets.


References:

[1] Beyond the 80/20 Rule: High-Entropy Minority Tokens Drive Effective Reinforcement Learning for LLM Reasoning

---

> ### Author Rebuttal · Authors · 2026-03-31
>
> We sincerely thank the reviewer for their thorough and constructive review. We are glad the reviewer found our work to be "the first to analyse CoT effectiveness via intrinsic dimensions" and "well-written and easy-to-understand." We address your raised points below.
>
> ---
>
> > \[W2\] Evaluation is limited to one dataset (GSM8K) and one model family (Gemma).
>
> We thank the reviewer and agree this point is well taken. To address this, we ran additional experiments on Reasoning Gym (Stojanovski et al., 2025\) across two categories: **cognitive reasoning** (color cube rotation, rectangle counting, number sequence) and **algorithmic reasoning** (leg counting, base conversion, matrix rotation), testing qualitatively different skills from GSM8K. We evaluate on in-distribution, hard in-distribution, and OOD conditions using Gemma-3 4B. Please refer to the response to **Reviewer eRSy** for the full setup and results. We will expand this to additional model families in the final version.
>
>
> ---
>
> > \[W3 / Q1\] ID is not an intrinsic characteristic of the model but triggered by datasets; training on different datasets may change ID for the same CoTs.
>
> The seminal work defining intrinsic dimensionality (Li et al., 2018\) explicitly identifies it as a property of the loss objective, i.e., the **(model, task) pair** jointly. The reviewer is correct that the absolute value of ID may change with a different problem distribution, and we acknowledge that the relative ordering of reasoning strategies could also differ across sufficiently different problem domains, where different reasoning strategies may be more or less effective. Crucially, however, we hypothesize that even across domains, intrinsic dimensionality will continue to correlate with generalization performance, as it captures a fundamental property of how compressible a reasoning strategy is for a given model and task.
>
> In support of this, our Reasoning Gym experiments (see response to **Reviewer eRSy**) provide direct evidence across two qualitatively different reasoning domains: cognitive and algorithmic reasoning, where the inverse correlation between intrinsic dimensionality and generalization holds consistently. As the problem distribution changes, so does the threshold τ (described in L67-85, L148-164, and Sec 4.3), keeping the measurement self-consistent across domains. We will expand on this analysis in the final version.
>
> ---
>
> >\[W1 / Q2\] How about PPL variants such as LongPPL?
>
> We thank the reviewer for the suggestion and include the results below. Using the same α and β values as the original LongPPL formulation with K=64 and K=256, we find Spearman correlations of 0.60 and 0.62 respectively on Gemma-3 4B.
>
>
> | Metric | Spearman Correlation |
> | :----------- | :---- |
> | LongPPL (K=64) | 0.60 |
> | LongPPL (K=256) | 0.62 |
> | Token Perplexity | 0.82 |
> | Intrinsic Dimensionality (ours) | 0.93 |
>
> Both LongPPL variants fall below regular token perplexity (0.82) and intrinsic dimensionality (0.93), which remains the strongest predictor of reasoning effectiveness. We note that LongPPL is primarily designed for long-context settings with a proposed K=4096 (length of short context), which comfortably covers all CoT strategies we evaluate. Token perplexity is already length-normalized and thus less sensitive to sequence length than KL divergence, which may explain why the additional long-context correction does not provide further gains in our setting.
>
> ---
>
> >\[W1 / Q3\] Faithfulness metrics; reference \[1\] should be covered.
>
> Your point is well taken. Regarding \[1\]: the key finding is that restricting gradient updates to high-entropy forking tokens improves performance. However, two distinctions are important.
>
> - First, this is an optimization technique rather than a predictive metric for comparing strategies.
> - Second, high-entropy tokens are tied to specific strategies, most prevalent in backtracking- and verification-heavy approaches like our High Review Ratio CoT baseline.
>
> We will include a detailed discussion of \[1\] in the next version. Regarding faithfulness metrics: prior work such as ROSCOE (Golovneva et al., 2023, already cited) primarily assesses instance-level correctness, complementary but distinct from what intrinsic dimensionality captures, namely the global learnability of a reasoning strategy across the full training distribution. We will expand our related work coverage in the final version.
>
> ---
>
> We hope these responses address the reviewer's concerns and allow them to revise their score in light of these clarifications, particularly given our new experiments. We are happy to address any remaining questions.

---

> > ### Author Rebuttal · Reviewer_pBeX · 2026-04-02
> >
> > Do we need to fine-tune the model on the extract dataset (for example, dataset A) for the CoT evaluation from dataset A? if so, how is the transferability and generalisability?

---

> > > ### Author Response · Authors · 2026-04-03
> > >
> > > We thank the reviewer for their follow-up question. We are happy to clarify the procedure and its implications for transferability and generalisability.
> > > Computing intrinsic dimensionality only requires access to CoTs over a **training set** -- we fine-tune LoRA adapters of varying ranks on the training split and identify the minimum parameter count needed to reach the accuracy threshold τ. Critically, this computation is performed entirely on the training split, and the resulting intrinsic dimensionality measurement is then used to **predict generalization** on both a held-out in-distribution validation set and out-of-distribution evaluations. The strong correlations we observe (0.93 on 4B, 0.75 on 1B) therefore reflect genuine predictive power on unseen data, not a post-hoc measurement on the same data used to compute ID. Our Reasoning Gym experiments (see response to **Reviewer eRSy**) further demonstrate that this predictive relationship holds across qualitatively different reasoning domains, providing direct evidence of transferability beyond GSM8K.
> > >
> > > We acknowledge that computing intrinsic dimensionality is computationally expensive, as it requires fine-tuning multiple LoRA adapters across a sweep of parameter counts. As noted in the last two sentences of our conclusion, an important direction for future work is identifying more efficiently computable approximations to intrinsic dimensionality, which would enable more practical applications such as efficiently selecting or optimizing CoT strategies at scale.
> > >
> > > We hope this clarifies the procedure and addresses any remaining questions.

---

### Official Review · Reviewer_eRSy · 2026-03-13

**Soundness:** 3
**Presentation:** 3
**Significance:** 3
**Originality:** 3
**Overall Recommendation:** 5
**Confidence:** 4

**Summary:**

This paper proposes intrinsic dimensionality (ID) operationalized as the minimum number of trainable parameters (via LoRA) required to reach a fixed training-accuracy threshold - as a quantitative metric for evaluating the effectiveness of different reasoning-chain strategies in language models. Keeping the model architecture fixed while varying task formulation through generated rationales, the authors observe a strong inverse correlation between ID and downstream generalization on GSM8K and multiple out-of-distribution (OOD) stress tests, across two model sizes (Gemma-3 1B and 4B) and a broad range of reasoning styles (e.g., short and long chain-of-thought, code execution, planning). They further show that ID outperforms commonly used proxies such as response length and token-level perplexity in predicting generalization, and they provide robustness analyses across different threshold choices.

**Compliance With Llm Reviewing Policy:**

Affirmed.

**Final Justification:**

Thank you to the authors for their response and experiments, I will keep my score.

**Key Questions For Authors:**

1. To what extent is the strong ID-generalization correlation specific to LoRA? Have you tried other PEFT methods (e.g., [1], [2]) or a random subspace approach to see if the correlation persists?

2. Can the authors provide results on more challenging mathematical reasoning datasets beyond GSM8K (e.g., MATH500, AIME 2024/2025)?

[1] : https://arxiv.org/abs/2104.08691
[2] : https://arxiv.org/abs/2402.09353

**Limitations:**

yes

**Strengths And Weaknesses:**

## Strengths
### Novelty
- Connects the effectiveness of reasoning chains to an information-theoretic lens (minimum description length) via intrinsic dimensionality of the learning objective.
- Adapts ID measurement to LLM fine-tuning by using LoRA subspace sweeps as structured low-dimensional projections; proposes a practical procedure to compare reasoning strategies with a common accuracy threshold.
- Frames a clear, testable hypothesis: more effective reasoning chains lower the intrinsic dimensionality required to learn the task mapping, hence generalize better.

### Presentation
- The core idea, measurement protocol, and thresholding procedure are clearly explained and supported by helpful figures.

### Experiment
- Evaluates 14 diverse reasoning strategies, spanning answer-only, short/long CoT, distractors, planning, and code-based execution


## Weaknesses
Evaluation is limited to one benchmark family (GSM8K and its variants). Demonstrating the pattern on additional reasoning datasets (e.g., MATH500) would strengthen generality.

---

> ### Author Rebuttal · Authors · 2026-03-31
>
> We sincerely thank the reviewer and are glad they found our work to have genuine novelty in "connecting the effectiveness of reasoning chains to an information-theoretic lens via intrinsic dimensionality," and appreciated that "the core idea, and measurement protocol are clearly explained and supported". Below we address your comments in detail.
>
> ---
>
> >\[Q1\] Is the ID-generalization correlation specific to LoRA? Would other PEFT methods or random subspace approaches yield similar results?
>
> We thank the reviewer for raising this important point. From a theoretical standpoint, the arguments in Li et al. (2018) and Aghajanyan et al. (2021) suggest that any layer-wise projection or PEFT method can provide an upper bound on intrinsic dimensionality, and should serve as an acceptable proxy for low-dimensional subspace training, including prompt and prefix tuning as you suggest. Prior work such as [Lotfi et al. (2024)](https://arxiv.org/abs/2312.17173) has also compared the compression ratios of LoRA, subspace training, and more complex combinations, which can provide even tighter bounds. However, our purpose is to make relative rather than absolute claims with respect to intrinsic dimensionality. So, we chose LoRA as our measurement protocol due to its widespread adoption and demonstrated effectiveness for LLM fine-tuning. Due to the time constraints of the rebuttal period, we were not able to empirically verify this across alternative PEFT methods, but we agree this is a valuable direction. We will add a detailed discussion of this point in the revision, and future work could further empirically validate whether the ordering provided by LoRA strongly correlates with that of other PEFT methods.
>
>
> ---
>
> >\[W1 / Q2\] Evaluation is limited to one benchmark family; results on harder benchmarks would strengthen generality.
>
> We note that MATH500 and AIME are test-only datasets and cannot be used in our pipeline. To address the spirit of this concern, we ran additional experiments on tasks from [Reasoning Gym](https://arxiv.org/abs/2505.24760) (Stojanovski et al., 2025), selecting two categories testing qualitatively different reasoning skills: **cognitive reasoning** (`color_cube_rotation`, `rectangle_count`, `number_sequence`) and **algorithmic reasoning** (`leg_counting`, `base_conversion`, `rotate_matrix`).
>
> Metrics are computed separately per group using Gemma-3 4B across: (i) an **ID** test set from easy configurations, (ii) a **hard OOD** test set using harder configurations of the same tasks, and (iii) an **OOD** test set of held-out tasks; for cognition: `needle_haystack` and `modulo_grid`; for algorithmic: `letter_counting` and `count_primes`.
>
> Due to the time constraints of the rebuttal window, we report results for a representative subset of reasoning strategies.
>
> **Algorithmic:**
>
> | CoT Strategy | ID | OOD | Overall | Intrinsic Dim. (M) ↓ | Token PPL ↓ | KL Div ↓ | Length ↑ |
> | :---- | :---: | :---: | :---: | :---: | :---: | :---: | :---: |
> | No CoT | 22.53 | 4.49 | 7.69 | 532.81 | 7.33 | 71.26 | 28.63 |
> | Short CoT | 51.60 | 11.40 | 18.86 | 3.92 | 1.78 | 105.75 | 140.88 |
> | Gemma 27B CoT | 57.71 | 11.68 | 19.90 | 1.49 | 1.20 | 39.44 | 191.79 |
> | Exec PoT | 96.67 | 63.71 | 73.21 | 0.74 | 1.28 | 52.43 | 176.52 |
> | Spearman Correl | \- | \- | \- | 1.0\* | 0.8\* | 0.6\* | 0.8\* |
>
> **Cognition:**
>
> | CoT Strategy | ID  | OOD | Overall | Intrinsic Dim. (M) ↓ | Token PPL ↓ | KL Div ↓ | Length ↑ |
> | :---- | :---: | :---: | :---: | :---: | :---: | :---: | :---: |
> | No CoT | 61.93 | 44.68 | 49.82 | 0.40 | 15.43 | 51.68 | 6.99 |
> | Short CoT | 51.85 | 37.99 | 42.15 | 2.05 | 5.75 | 145.79 | 62.12 |
> | Gemma 27B CoT | 44.87 | 36.06 | 38.78 | 3.92 | 1.60 | 113.79 | 182.89 |
> | Exec PoT | 51.14 | 35.04 | 39.74 | 2.55 | 1.61 | 112.41 | 178.67 |
> | Spearman Correl | \- | \- | \- | 1.0\* | \-1.0\* | 0.4\* | \-1.0\* |
>
> *Metrics marked with ↓ indicate lower values predict higher accuracy. \*Comparing only 4 strategies; not statistically significant. Reported for illustrative purposes only.*
>
> Key takeaways:
>
> * **Algorithmic:** Exec PoT achieves the lowest intrinsic dimensionality (0.74M) and best overall performance (73.21). No CoT requires the most capacity (532.81M) and performs worst, mirroring our GSM8K findings.
> * **Cognition:** No CoT achieves the lowest intrinsic dimensionality (0.40M) and best overall performance (49.82), consistent with our framework – pattern recognition tasks are more compressible without intermediate reasoning steps, consistent with Sprague et al. (2025).
> * **KL divergence and length** show weaker and less consistent relationships with performance across both categories, further supporting intrinsic dimensionality as the most reliable predictor.
>
> We will include these results in the next version of our work. Please let us know if you have any other questions.

---

> > ### Author Rebuttal · Reviewer_eRSy · 2026-04-02
> >
> > Thank you to the authors for their response and experiments, I will keep my score.

---

> > > ### Author Response · Authors · 2026-04-07
> > >
> > > Thank you for your positive assessment and continued engagement. We will incorporate our additional experiments and responses in the final version.

---

### Decision · Program_Chairs · 2026-04-30

**Decision:**

Accept (spotlight)

**Comment:**

The paper proposes intrinsic dimensionality as a quantitative measure for analyzing the effectiveness of reasoning-chain strategies. The authors operationalize intrinsic dimensionality as the minimum number of trainable LoRA parameters required for a fixed pretrained model to reach a common training-accuracy threshold under a given reasoning strategy. By keeping the base model fixed and varying the training targets across different reasoning strategies, the paper studies whether reasoning chains that are easier to fit in a low-dimensional adaptation subspace also generalize better.

The reviewers generally found the idea interesting and the experimental design clear. The main empirical finding is that, across 14 reasoning strategies on GSM8K with Gemma-3 1B and 4B models, lower intrinsic dimensionality is strongly correlated with better in-distribution and out-of-distribution generalization. The correlation is stronger than several simpler proxies, including response length, token perplexity, and sequence-level KL divergence, and the result appears robust to the threshold used to define intrinsic dimensionality. Reviewers also appreciated that the method offers a concrete and testable way to compare reasoning strategies rather than relying only on qualitative claims about structure, length, or test-time computation.

Some limitations remain and should be clearly reflected in the final manuscript. The evidence supports intrinsic dimensionality as a useful quantitative predictor of reasoning-strategy effectiveness, but it should not be overstated as a fully established causal mechanism. The main measurement protocol is also expensive and may depend on the choice of PEFT parameterization, so future work is needed to find cheaper proxies and to verify the ordering across broader models, tasks, and adaptation methods. Nevertheless, the paper presents a novel and well-motivated analysis framework with strong empirical correlations and a consistently positive reviewer consensus. I therefore recommend acceptance.